# Framing RNN as a kernel method:
# A neural ODE approach

**Adeline Fermanian**[1*]     **Pierre Marion**[1*]     **Jean-Philippe Vert**[2]     **Gérard Biau**[1]

[1] Sorbonne Université, CNRS,
Laboratoire de Probabilités, Statistique et Modélisation, LPSM,
F-75005 Paris, France
{adeline.fermanian, pierre.marion, gerard.biau}@sorbonne-universite.fr
[2] Google Research, Brain team,
Paris, France
jpvert@google.com

## Abstract

Building on the interpretation of a recurrent neural network (RNN) as a continuous-time neural differential equation, we show, under appropriate conditions, that the solution of a RNN can be viewed as a linear function of a specific feature set of the input sequence, known as the signature. This connection allows us to frame a RNN as a kernel method in a suitable reproducing kernel Hilbert space. As a consequence, we obtain theoretical guarantees on generalization and stability for a large class of recurrent networks. Our results are illustrated on simulated datasets.

## 1  Introduction

Recurrent neural networks (RNN) are among the most successful methods for modeling sequential data. They have achieved state-of-the-art results in difficult problems such as natural language processing (e.g., Mikolov et al., 2010; Collobert et al., 2011) or speech recognition (e.g., Hinton et al., 2012; Graves et al., 2013). This class of neural networks has a natural interpretation in terms of (discretization of) ordinary differential equations (ODE), which casts them in the field of neural ODE (Chen et al., 2018). This observation has led to the development of continuous-depth models for handling irregularly-sampled time-series data, including the ODE-RNN model (Rubanova et al., 2019), GRU-ODE-Bayes (De Brouwer et al., 2019), or neural CDE models (Kidger et al., 2020; Morrill et al., 2020a). In addition, the time-continuous interpretation of RNN allows to leverage the rich theory of differential equations to develop new recurrent architectures (Chang et al., 2019; Herrera et al., 2020; Erichson et al., 2021), which are better at learning long-term dependencies.

On the other hand, the development of kernel methods for deep learning offers theoretical insights on the functions learned by the networks (Cho and Saul, 2009; Belkin et al., 2018; Jacot et al., 2018). Here, the general principle consists in defining a reproducing kernel Hilbert space (RKHS)—that is, a function class $\mathscr{H}$—, which is rich enough to describe the architectures of networks. A good example is the construction of Bietti and Mairal (2017, 2019), who exhibit an RKHS for convolutional neural networks. This kernel perspective has several advantages. First, by separating the representation of the data from the learning process, it allows to study invariances of the representations learned by the network. Next, by reducing the learning problem to a linear one in $\mathscr{H}$, generalization bounds can be more easily obtained. Finally, the Hilbert structure of $\mathscr{H}$ provides a natural metric on neural networks, which can be used for example for regularization (Bietti et al., 2019).

---

*Equal contribution

35th Conference on Neural Information Processing Systems (NeurIPS 2021).

**Contributions.**    By taking advantage of the neural ODE paradigm for RNN, we show that RNN are, in the continuous-time limit, linear predictors over a specific space associated with the signature of the input sequence (Levin et al., 2013). The signature transform, first defined by Chen (1958) and central in rough path theory (Lyons et al., 2007; Friz and Victoir, 2010), summarizes sequential inputs by a graded feature set of their iterated integrals. Its natural environment is a tensor space that can be endowed with an RKHS structure (Király and Oberhauser, 2019). We exhibit general conditions under which classical recurrent architectures such as feedforward RNN, Gated Recurrent Units (GRU, Cho et al., 2014), or Long Short-Term Memory networks (LSTM, Hochreiter and Schmidhuber, 1997), can be framed as a kernel method in this RKHS. This enables us to provide generalization bounds for RNN as well as stability guarantees via regularization. The theory is illustrated with some experimental results.

**Related works.**    The neural ODE paradigm was first formulated by Chen et al. (2018) for residual neural networks. It was then extended to RNN in several articles, with a focus on handling irregularly sampled data (Rubanova et al., 2019; Kidger et al., 2020) and learning long-term dependencies (Chang et al., 2019). The signature transform has recently received the attention of the machine learning community (Levin et al., 2013; Kidger et al., 2019; Liao et al., 2019; Toth and Oberhauser, 2020; Fermanian, 2021) and, combined with deep neural networks, has achieved state-of-the-art performance for several applications (Yang et al., 2016, 2017; Perez Arribas, 2018; Wang et al., 2019; Morrill et al., 2020b). Király and Oberhauser (2019) use the signature transform to define kernels for sequential data and develop fast computational methods. The connection between continuous-time RNN and signatures has been pointed out by Lim (2021) for a specific model of stochastic RNN. Deriving generalization bounds for RNN is an active research area (Zhang et al., 2018; Akpinar et al., 2019; Tu et al., 2019). By leveraging the theory of differential equations, our approach encompasses a large class of RNN models, ranging from feedforward RNN to LSTM. This is in contrast with most existing generalization bounds, which are architecture-dependent. Close to our point of view is the work of Bietti and Mairal (2017) for convolutional neural networks.

**Mathematical context.**    We place ourselves in a supervised learning setting. The input data is a sample of $n$ i.i.d. vector-valued sequences $\{\mathbf{x}^{(1)}, \ldots, \mathbf{x}^{(n)}\}$, where $\mathbf{x}^{(i)} = (x_1^{(i)}, \ldots, x_T^{(i)}) \in (\mathbb{R}^d)^T$, $T \geq 1$. The outputs of the learning problem can be either labels (classification setting) or sequences (sequence-to-sequence setting). Even if we only observe discrete sequences, each $\mathbf{x}^{(i)}$ is mathematically considered as a regular discretization of a continuous-time process $X^{(i)} \in BV^c([0,1], \mathbb{R}^d)$, where $BV^c([0,1], \mathbb{R}^d)$ is the space of continuous functions from $[0,1]$ to $\mathbb{R}^d$ of finite total variation. Informally, the total variation of a process corresponds to its length. Formally, for any $[s,t] \subset [0,1]$, the total variation of a process $X \in BV^c([0,1], \mathbb{R}^d)$ on $[s,t]$ is defined by

$$\|X\|_{TV;[s,t]} = \sup_{(t_0,\ldots,t_k) \in D_{s,t}} \sum_{j=1}^{k} \|X_{t_j} - X_{t_{j-1}}\|,$$

where $D_{s,t}$ denotes the set of all finite partitions of $[s,t]$ and $\|\cdot\|$ the Euclidean norm. We therefore have that $x_j^{(i)} = X_{j/T}^{(i)}$, $1 \leq j \leq T$, where $X_t^{(i)} := X^{(i)}(t)$. We make two assumptions on the processes $X^{(i)}$. First, they all begin at zero, and second, their lengths are bounded by $L \in (0,1)$. These assumptions are not too restrictive, since they amount to data translation and normalization, common in practice. Accordingly, we denote by $\mathscr{X}$ the subset of $BV^c([0,1], \mathbb{R}^d)$ defined by

$$\mathscr{X} = \left\{ X \in BV^c([0,1], \mathbb{R}^d) \,|\, X_0 = 0 \quad \text{and} \quad \|X\|_{TV;[0,1]} \leq L \right\}$$

and assume therefore that $X^{(1)}, \ldots, X^{(n)}$ are i.i.d. according to some $X \in \mathscr{X}$. The norm on all spaces $\mathbb{R}^m$, $m \geq 1$, is always the Euclidean one. Observe that assuming that $X \in \mathscr{X}$ implies that, for any $t \in [0,1]$, $\|X_t\| = \|X_t - X_0\| \leq \|X\|_{TV;[0,1]} \leq L$.

**Recurrent neural networks.**    Classical RNN are defined by a sequence of hidden states $h_1, \ldots, h_T \in \mathbb{R}^e$, where, for $\mathbf{x} = (x_1, \ldots, x_T)$ a generic data sample,

$$h_0 = 0 \quad \text{and} \quad h_{j+1} = f(h_j, x_{j+1}) \quad \text{for } 0 \leq j \leq T - 1.$$

At each time step $1 \leq j \leq T$, the output of the network is $z_j = \psi(h_j)$, where $\psi$ is a linear function. In the present article, we rather consider the following residual version, which is a natural adaptation

of classical RNN in the neural ODE framework (see, e.g., Yue et al., 2018):

$$h_0 = 0 \quad \text{and} \quad h_{j+1} = h_j + \frac{1}{T}f(h_j, x_{j+1}) \quad \text{for } 0 \leq j \leq T - 1. \tag{1}$$

The simplest choice for the function $f$ is the feedforward model, say $f_{\text{RNN}}$, defined by

$$f_{\text{RNN}}(h, x) = \sigma(Uh + Vx + b), \tag{2}$$

where $\sigma$ is an activation function, $U \in \mathbb{R}^{e \times e}$ and $V \in \mathbb{R}^{e \times d}$ are weight matrices, and $b \in \mathbb{R}^e$ is the bias. The function $f_{\text{RNN}}$, equipped with a smooth activation $\sigma$ (such as the logistic or hyperbolic tangent functions), will be our leading example throughout the paper. However, the GRU and LSTM models can also be rewritten under the form (1), as shown in Appendix A.1. Thus, model (1) is flexible enough to encompass most recurrent networks used in practice.

**Overview.** Section 2 is devoted to framing RNN as linear functions in a suitable RKHS. We start by embedding iteration (1) into a continuous-time model, which takes the form of a controlled differential equation (CDE). This allows, after introducing the signature transform, to define the appropriate RKHS, and, in turn, to show that model (1) boils down, in the continuous-time limit, to a linear problem on the signature. This framework is used in Section 3 to derive generalization bounds and stability guarantees. We provide some experiments in Section 4 before discussing our results in Section 5. All proofs are postponed to the supplementary material.

## 2 Framing RNN as a kernel method

**Roadmap.** First, we quantify the difference between the discrete recurrent network (1) and its continuous-time counterpart (Proposition 1). Then, we rewrite the corresponding ODE as a CDE (Proposition 2). Under appropriate conditions, Proposition 4 shows that the solution of this equation is a linear function of the signature of the driving process. Importantly, these assumptions are valid for a feedforward RNN, as stated by Proposition 5. We conclude in Theorem 1.

### 2.1 From discrete to continuous time

Recall that $h_0, \ldots, h_T$ denote the hidden states of the RNN (1), and let $H : [0, 1] \to \mathbb{R}^e$ be the solution of the ODE

$$dH_t = f(H_t, X_t)dt, \quad H_0 = h_0. \tag{3}$$

By bounding the difference between $H_{j/T}$ and $h_j$, the following proposition shows how to pass from discrete to continuous time, provided $f$ satisfies the following assumption:

($A_1$)  The function $f$ is Lipschitz continuous in $h$ and $x$, with Lipschitz constants $K_h$ and $K_x$. We let $K_f = \max(K_h, K_x)$.

**Proposition 1.** *Assume that ($A_1$) is verified. Then there exists a unique solution $H$ to (3) and, for any $0 \leq j \leq T$,*

$$\|H_{j/T} - h_j\| \leq \frac{c_1}{T},$$

*where $c_1 = K_f e^{K_f}\big(L + \sup\limits_{\|h\| \leq M, \|x\| \leq L} \|f(h, x)\|e^{K_f}\big)$ and $M = \sup\limits_{\|x\| \leq L} \|f(h_0, x)\|e^{K_f}$. Moreover, for any $t \in [0, 1]$, $\|H_t\| \leq M$.*

Then, following Kidger et al. (2020), we show that the ODE (3) can be rewritten under the form of a CDE. At the cost of increasing the dimension of the hidden state from $e$ to $e + d$, this allows us to reframe model (3) as a linear model in $dX$, in the sense that $X$ has been moved 'outside' of $f$.

**Proposition 2.** *Assume that ($A_1$) is verified. Let $H : [0, 1] \to \mathbb{R}^e$ be the solution of (3), and let $\bar{X} : [0, 1] \to \mathbb{R}^{d+1}$ be the time-augmented process $\bar{X}_t = (X_t^\top, \frac{1-L}{2}t)^\top$. Then there exists a tensor field $\mathbf{F} : \mathbb{R}^{\bar{e}} \to \mathbb{R}^{\bar{e} \times \bar{d}}, \bar{e} = e + d, \bar{d} = d + 1$, such that if $\bar{H} : [0, 1] \to \mathbb{R}^{\bar{e}}$ is the solution of the CDE*

$$d\bar{H}_t = \mathbf{F}(\bar{H}_t)d\bar{X}_t, \quad \bar{H}_0 = (H_0^\top, X_0^\top)^\top, \tag{4}$$

*then its first $e$ coordinates are equal to $H$.*

Equation (4) can be better understood by the following equivalent integral equation:

$$\bar{H}_t = \bar{H}_0 + \int_0^t \mathbf{F}(\bar{H}_u)d\bar{X}_u,$$

where the integral should be understood as Riemann-Stieljes integral (Friz and Victoir, 2010, Section I.2). Thus, the output of the RNN can be approximated by the solution of the CDE (4), and, according to Proposition 1, the approximation error is $\mathcal{O}(1/T)$.

**Example 1.** *Consider $f_{\mathrm{RNN}}$ as in (2). If $\sigma$ is Lipschitz continuous with constant $K_\sigma$, then, for any $h_1, h_2 \in \mathbb{R}^e$, $x_1, x_2 \in \mathbb{R}^d$,*

$$\|f_{\mathrm{RNN}}(h_1, x_1) - f_{\mathrm{RNN}}(h_2, x_1)\| = \|\sigma(Uh_1 + Vx_1 + b) - \sigma(Uh_2 + Vx_1 + b)\|$$
$$\leq K_\sigma \|U\|_{\mathrm{op}}\|h_1 - h_2\|,$$

*where $\|\cdot\|_{\mathrm{op}}$ denotes the operator norm—see Appendix A.3. Similarly, $\|f(h_1, x_1) - f(h_1, x_2)\| \leq K_\sigma \|V\|_{\mathrm{op}}\|x_1 - x_2\|$. Thus, assumption $(A_1)$ is satisfied. The tensor field $\mathbf{F}_{\mathrm{RNN}}$ of Proposition 2 corresponding to this network is defined for any $\bar{h} \in \mathbb{R}^{\bar{e}}$ by*

$$\mathbf{F}_{\mathrm{RNN}}(\bar{h}) = \begin{pmatrix} 0_{e \times d} & \frac{2}{1-L}\sigma(W\bar{h} + b) \\ I_{d \times d} & 0_{d \times 1} \end{pmatrix}, \quad where \quad W = (U \quad V) \in \mathbb{R}^{e \times \bar{e}}. \tag{5}$$

## 2.2  The signature

An essential ingredient towards our construction is the signature of a continuous-time process, which we briefly present here. We refer to Chevyrev and Kormilitzin (2016) for a gentle introduction and to Lyons et al. (2007); Levin et al. (2013) for details.

**Tensor Hilbert spaces.**   We denote by $(\mathbb{R}^d)^{\otimes k}$ the $k$th tensor power of $\mathbb{R}^d$ with itself, which is a Hilbert space of dimension $d^k$. The key space to define the signature and, in turn, our RKHS, consists in infinite square-summable sequences of tensors of increasing order:

$$\mathscr{T} = \left\{ a = (a_0, \dots, a_k, \dots) \,\Big|\, a_k \in (\mathbb{R}^d)^{\otimes k}, \sum_{k=0}^{\infty} \|a_k\|_{(\mathbb{R}^d)^{\otimes k}}^2 < \infty \right\}. \tag{6}$$

Endowed with the scalar product $\langle a, b \rangle_{\mathscr{T}} := \sum_{k=0}^{\infty} \langle a_k, b_k \rangle_{(\mathbb{R}^d)^{\otimes k}}$, $\mathscr{T}$ is a Hilbert space, as shown in Appendix A.4.

**Definition 1.** *Let $X \in BV^c([0,1], \mathbb{R}^d)$. For any $t \in [0,1]$, the signature of $X$ on $[0,t]$ is defined by $S_{[0,t]}(X) = (1, \mathbb{X}_{[0,t]}^1, \dots, \mathbb{X}_{[0,t]}^k, \dots)$, where, for each $k \geq 1$,*

$$\mathbb{X}_{[0,t]}^k = k! \int \cdots \int_{0 \leq u_1 < \cdots < u_k \leq t} dX_{u_1} \otimes \cdots \otimes dX_{u_k} \in (\mathbb{R}^d)^{\otimes k}.$$

Although this definition is technical, the signature should simply be thought of as a feature map that embeds a bounded variation process into an infinite-dimensional tensor space. The signature has several good properties that make it a relevant tool for machine learning (e.g., Levin et al., 2013; Chevyrev and Kormilitzin, 2016; Fermanian, 2021). In particular, under certain assumptions, $S(X)$ characterizes $X$ up to translations and reparameterizations, and has good approximation properties. We also highlight that fast libraries exist for computing the signature (Reizenstein and Graham, 2020; Kidger and Lyons, 2021).

The expert reader is warned that this definition differs from the usual one by the normalization of $\mathbb{X}_{[0,t]}^k$ by $k!$, which is more adapted to our context. In the sequel, for any index $(i_1, \dots, i_k) \subset \{1, \dots, d\}^k$, $S_{[0,t]}^{(i_1, \dots, i_k)}(X)$ denotes the term associated with the coordinates $(i_1, \dots, i_k)$ of $\mathbb{X}_{[0,t]}^k$. When the signature is taken on the whole interval $[0,1]$, we simply write $S(X)$, $S^{(i_1, \dots, i_k)}(X)$, and $\mathbb{X}^k$.

**Example 2.** *Let $X$ be the $d$-dimensional linear path defined by $X_t = (a_1 + b_1 t, \dots, a_d + b_d t)^\top$, $a_i, b_i \in \mathbb{R}$. Then $S^{(i_1, \dots, i_k)}(X) = b_{i_1} \cdots b_{i_k}$ and $\mathbb{X}^k = b^{\otimes k}$.*

The next proposition, which ensures that $S_{[0,t]}(\bar{X}) \in \mathscr{T}$, is an important step.

**Proposition 3.** *Let $X \in \mathscr{X}$ and $\bar{X}_t = (X_t^\top, \frac{1-L}{2}t)^\top$ as in Proposition 2. Then, for any $t \in [0,1]$, $\|S_{[0,t]}(\bar{X})\|_{\mathscr{T}} \leq 2(1-L)^{-1}$.*

**The signature kernel.** By taking advantage of the structure of Hilbert space of $\mathscr{T}$, it is natural to introduce the following kernel:

$$K : \mathscr{X} \times \mathscr{X} \to \mathbb{R}$$
$$(X, Y) \mapsto \langle S(\bar{X}), S(\bar{Y}) \rangle_{\mathscr{T}},$$

which is well defined according to Proposition 3. We refer to Király and Oberhauser (2019) for a general presentation of kernel methods with signatures and to Cass et al. (2020) for a kernel trick. The RKHS associated with $K$ is the space of functions

$$\mathscr{H} = \left\{ \xi_\alpha : \mathscr{X} \to \mathbb{R} \,|\, \xi_\alpha(X) = \langle \alpha, S(\bar{X}) \rangle_{\mathscr{T}}, \alpha \in \mathscr{T} \right\}, \tag{7}$$

with scalar product $\langle \xi_\alpha, \xi_\beta \rangle_{\mathscr{H}} = \langle \alpha, \beta \rangle_{\mathscr{T}}$ (see, e.g., Schölkopf and Smola, 2002).

## 2.3 From the CDE to the signature kernel

An important property of signatures is that the solution of the CDE (4) can be written, under certain assumptions, as a linear function of the signature of the driving process $X$. This operation can be thought of as a Taylor expansion for CDE. More precisely, let us rewrite (4) as

$$dH_t = \mathbf{F}(H_t)dX_t = \sum_{i=1}^{d} F^i(H_t)dX_t^i, \tag{8}$$

where $X_t = (X_t^1, \ldots, X_t^d)^\top$, $\mathbf{F} : \mathbb{R}^e \to \mathbb{R}^{e \times d}$, and $F^i : \mathbb{R}^e \to \mathbb{R}^e$ are the columns of $\mathbf{F}$—to avoid heavy notation, we momentarily write $e$, $d$, $H$, and $X$ instead of $\bar{e}$, $\bar{d}$, $\bar{H}$, and $\bar{X}$. Throughout, the bold notation is used to distinguish tensor fields and vector fields. We recall that a vector field $F : \mathbb{R}^e \to \mathbb{R}^e$ or a tensor field $\mathbf{F} : \mathbb{R}^e \to \mathbb{R}^{e \times d}$ are said to be smooth if each of their coordinates is $\mathscr{C}^\infty$.

**Definition 2.** *Let $F, G : \mathbb{R}^e \to \mathbb{R}^e$ be smooth vector fields and denote by $J(\cdot)$ the Jacobian matrix. Their differential product is the smooth vector field $F \star G : \mathbb{R}^e \to \mathbb{R}^e$ defined, for any $h \in \mathbb{R}^e$, by*

$$(F \star G)(h) = \sum_{j=1}^{e} \frac{\partial G}{\partial h_j}(h) F_j(h) = J(G)(h)F(h).$$

In differential geometry, $F \star G$ is simply denoted by $FG$. Since the $\star$ operation is not associative, we take the convention that it is evaluated from right to left, i.e., $F^1 \star F^2 \star F^3 := F^1 \star (F^2 \star F^3)$.

**Taylor expansion.** Let $H$ be the solution of (8), where $\mathbf{F}$ is assumed to be smooth. We now show that $H$ can be written as a linear function of the signature of $X$, which is the crucial step to embed the RNN in the RKHS $\mathscr{H}$. The step-$N$ Taylor expansion of $H$ (Friz and Victoir, 2008) is defined by

$$H_t^N = H_0 + \sum_{k=1}^{N} \frac{1}{k!} \sum_{1 \le i_1, \ldots, i_k \le d} S_{[0,t]}^{(i_1, \ldots, i_k)}(X) F^{i_1} \star \cdots \star F^{i_k}(H_0).$$

Throughout, we let

$$\Lambda_k(\mathbf{F}) = \sup_{\|h\| \le M, 1 \le i_1, \ldots, i_k \le d} \|F^{i_1} \star \cdots \star F^{i_k}(h)\|.$$

**Example 3.** *Let $\mathbf{F} = \mathbf{F}_{\mathrm{RNN}}$ defined by (5) with an identity activation. Then, for any $\bar{h} \in \mathbb{R}^{\bar{e}}$, $1 \le i \le d + 1$, $F_{\mathrm{RNN}}^i(\bar{h}) = W_i\bar{h} + b_i$, where $b_i$ is the $(i + d)$th vector of the canonical basis of $\mathbb{R}^{\bar{e}}$, and*

$$W_i = 0_{\bar{e} \times \bar{e}}, \quad W_{d+1} = \begin{pmatrix} \frac{2}{1-L}W \\ 0_{d \times \bar{e}} \end{pmatrix}, \quad \text{and} \quad b_{d+1} = \begin{pmatrix} \frac{2}{1-L}b \\ 0_d \end{pmatrix}.$$

*The vector fields $F_{\mathrm{RNN}}^i$ are then affine, $J(F_{\mathrm{RNN}}^i) = W_i$, and the iterated star products have a simple expression: for any $1 \le i_1, \ldots, i_k \le d$, $F_{\mathrm{RNN}}^{i_1} \star \cdots \star F_{\mathrm{RNN}}^{i_k}(\bar{h}) = W_{i_k} \cdots W_{i_2}(W_{i_1}\bar{h} + b_{i_1})$.*

The next proposition shows that the step-$N$ Taylor expansion $H^N$ is a good approximation of $H$.

**Proposition 4.** *Assume that the tensor field $\mathbf{F}$ is smooth. Then, for any $t \in [0, 1]$, $N \ge 1$,*

$$\|H_t - H_t^N\| \le \frac{d^{N+1}}{(N+1)!}\Lambda_{N+1}(\mathbf{F}). \tag{9}$$

Thus, provided that $\Lambda_N(\mathbf{F})$ is not too large, the right-hand side of (9) converges to zero, hence

$$H_t = H_0 + \sum_{k=1}^{\infty} \frac{1}{k!} \sum_{1 \leq i_1, \ldots, i_k \leq d} S_{[0,t]}^{(i_1, \ldots, i_k)}(X) F^{i_1} \star \cdots \star F^{i_k}(H_0). \tag{10}$$

We conclude from the above representation that the solution $H$ of (8) is in fact a linear function of the signature of $X$. A natural concern is to know whether the upper bound of Proposition 4 vanishes with $N$ for standard architectures. This property is encapsulated in the following more general assumption:

$$(A_2) \quad \text{The tensor field } \mathbf{F} \text{ is smooth and } \sum_{k=0}^{\infty} \left( \frac{d^k}{k!} \Lambda_k(\mathbf{F}) \right)^2 < \infty.$$

Clearly, if $(A_2)$ is verified, then the right-hand side of (9) converges to 0. The next proposition states formally the conditions under which $(A_2)$ is verified for $\mathbf{F}_{\text{RNN}}$. It is further illustrated in Figure 1, which shows that the convergence is fast with two common activation functions. We let $\|\sigma\|_\infty = \sup_{\|h\| \leq M, \|x\| \leq L} \|\sigma(Uh + Vx + b)\|$ and $\|\sigma^{(k)}\|_\infty = \sup_{\|h\| \leq M, \|x\| \leq L} \|\sigma^{(k)}(Uh + Vx + b)\|$, where $\sigma^{(k)}$ is the derivative of order $k$ of $\sigma$.

**Proposition 5.** *Let $\mathbf{F}_{RNN}$ be defined by (5). If $\sigma$ is the identity function, then $(A_2)$ is satisfied. In the general case, $(A_2)$ holds if $\sigma$ is smooth and there exists $a > 0$ such that, for any $k \geq 0$,*

$$\|\sigma^{(k)}\|_\infty \leq a^{k+1} k! \quad \text{and} \quad \|W\|_F < \frac{1-L}{8a^2 d}, \tag{11}$$

*where $\| \cdot \|_F$ is the Frobenius norm. Moreover, for any $N \geq 1$,*

$$\Lambda_N(\mathbf{F}_{RNN}) \leq \sqrt{2} a \left( \frac{8a^2 \|W\|_F}{1-L} \right)^{N-1} N! \,.$$

The proof of Proposition 5, based on the manipulation of higher-order derivatives of tensor fields, is highly non-trivial. We highlight that the conditions on $\sigma$ are mild and verified for common smooth activations. For example, they are verified for the logistic function (with $a = 2$) and for the hyperbolic tangent function (with $a = 4$)—see Appendix A.5. The second inequality of (11) puts a constraint on the norm of the weights, and can be regarded as a radius of convergence for the Taylor expansion.

**Putting everything together.** We now have all the elements at hand to embed the RNN into the RKHS $\mathscr{H}$. To fix the idea, we assume in this paragraph that we are in a $\pm 1$ classification setting. In other words, given an input sequence $\mathbf{x}$, we are interested in the final output $z_T = \psi(h_T) \in \mathbb{R}$, where $h_T$ is the solution of (1). The predicted class is $2 \cdot \mathbf{1}(z_T > 0) - 1$.

By Propositions 1 and 2, $z_T$ is approximated by the first $e$ coordinates of the solution of the CDE (4), which outputs a $\mathbb{R}^{e+d}$-valued process $\bar{H}$. According to Proposition 4, $\bar{H}$ is a linear function of the signature of the time-augmented process $\bar{X}$. Thus, on top of $\bar{H}$, it remains to successively apply the projection Proj on the $e$ first coordinates followed by the linear function $\psi$ to obtain an element of the RKHS $\mathscr{H}$. This mechanism is summarized in the following theorem.

**Theorem 1.** *Assume that $(A_1)$ and $(A_2)$ are verified. Then there exists a function $\xi_\alpha \in \mathscr{H}$ such that*

$$|z_T - \xi_\alpha(X)| \leq \|\psi\|_{\text{op}} \frac{c_1}{T}, \tag{12}$$

*where $\xi_\alpha(X) = \langle \alpha, S(\bar{X}) \rangle_{\mathscr{T}}$ and $\bar{X}_t = (X_t^\top, \frac{1-L}{2} t)^\top$. We have $\alpha = (\alpha_k)_{k=0}^\infty$, where each $\alpha_k \in (\mathbb{R}^d)^{\otimes k}$ is defined by*

$$\alpha_k^{(i_1, \ldots, i_k)} = \frac{1}{k!} \psi \circ \text{Proj}\big( F^{i_1} \star \cdots \star F^{i_k}(\bar{H}_0) \big).$$

*Moreover, $\|\alpha\|_{\mathscr{T}}^2 \leq \|\psi\|_{\text{op}}^2 \sum_{k=0}^{\infty} \left( \frac{d^k}{k!} \Lambda_k(\mathbf{F}) \right)^2$.*

We conclude that in the continuous-time limit, the output of the network can be interpreted as a scalar product between the signature of the (time-augmented) process $\bar{X}$ and an element of $\mathscr{T}$. This interpretation is important for at least two reasons: $(i)$ it facilitates the analysis of generalization of RNN by leveraging the theory of kernel methods, and $(ii)$ it provides new insights on regularization strategies to make RNN more robust. These points will be explored in the next section. Finally, we stress that the approach works for a large class of RNN, such as GRU and LSTM. The derivation of conditions $(A_1)$ and $(A_2)$ beyond the feedforward RNN is left for future work.

# 3 Generalization and regularization

## 3.1 Generalization bounds

**Learning procedure.** A first consequence of framing a RNN as a kernel method is that it gives natural generalization bounds under mild assumptions. In the learning setup, we are given an i.i.d. sample $\mathscr{D}_n$ of $n$ random pairs of observations $(\mathbf{x}^{(i)}, \mathbf{y}^{(i)}) \in (\mathbb{R}^d)^T \times \mathscr{Y}$, where $\mathbf{x}^{(i)} = (x_1^{(i)}, \ldots, x_T^{(i)})$. We distinguish the binary classification problem, where $\mathscr{Y} = \{-1, 1\}$, from the sequential prediction problem, where $\mathscr{Y} = (\mathbb{R}^p)^T$ and $\mathbf{y}^{(i)} = (y_1^{(i)}, \ldots, y_T^{(i)})$. The RNN is assumed to be parameterized by $\theta \in \Theta \subset \mathbb{R}^q$, where $\Theta$ is a compact set. To clarify the notation, we use a $\theta$ subscript whenever a quantity depends on $\theta$ (e.g., $f_\theta$ for $f$, etc.). In line with Section 2, it is assumed that the tensor field $\mathbf{F}_\theta$ associated with $f_\theta$ satisfies $(A_1)$ and $(A_2)$, keeping in mind that Proposition 5 guarantees that these requirements are fulfilled by a feedforward recurrent network with a smooth activation function.

Let $g_\theta : (\mathbb{R}^d)^T \to \mathscr{Y}$ denote the output of the recurrent network. The parameter $\theta$ is fitted by empirical risk minimization using a loss function $\ell : \mathscr{Y} \times \mathscr{Y} \to \mathbb{R}^+$. The theoretical and empirical risks are respectively defined, for any $\theta \in \Theta$, by

$$\mathscr{R}(\theta) = \mathbb{E}[\ell(\mathbf{y}, g_\theta(\mathbf{x}))] \quad \text{and} \quad \widehat{\mathscr{R}}_n(\theta) = \frac{1}{n}\sum_{i=1}^n \ell\big(\mathbf{y}^{(i)}, g_\theta(\mathbf{x}^{(i)})\big),$$

where the expectation $\mathbb{E}$ is evaluated with respect to the distribution of the generic random pair $(\mathbf{x}, \mathbf{y})$. We let $\widehat{\theta}_n \in \operatorname{argmin}_{\theta \in \Theta} \widehat{\mathscr{R}}_n(\theta)$ and aim at upper bounding $\mathbb{P}\big(\mathbf{y} g_{\widehat{\theta}_n}(\mathbf{x}) \leq 0 | \mathscr{D}_n\big)$ in the classification regime (Theorem 2) and $\mathscr{R}(\widehat{\theta}_n)$ in the sequential regime (Theorem 3). To reach this goal, our strategy is to approximate the RNN by its continuous version and then use the RKHS machinery of Section 2.

**Binary classification.** In this context, the network outputs a real number $g_\theta(\mathbf{x}) = \psi(h_T) \in \mathbb{R}$ and the predicted class is $2 \cdot \mathbf{1}(g_\theta(\mathbf{x}) > 0) - 1$. The loss $\ell : \{-1, 1\} \times \mathbb{R} \to \mathbb{R}^+$ is assumed to satisfy the assumptions of Bartlett and Mendelson (2002, Theorem 7), that is, for any $y \in \{-1, 1\}$, $z \in \mathbb{R}$, $\ell(y, z) = \phi(yz)$, where $\phi(u) \geq \mathbf{1}(u \leq 0)$, and $\phi$ is Lipschitz-continuous with constant $K_\ell$. For example, the logistic loss satisfies such assumptions. We let $\xi_{\alpha_\theta} \in \mathscr{H}$ be the function of Theorem 1 that approximates the RNN with parameter $\theta$. Thus, $z_T \approx \xi_{\alpha_\theta}(\bar{X}) = \langle \alpha_\theta, S(\bar{X}) \rangle_{\mathscr{T}}$, up to a $\mathscr{O}(1/T)$ term.

**Theorem 2.** *Assume that for all $\theta \in \Theta$, $(A_1)$ and $(A_2)$ are verified. Assume, in addition, that there exists a constant $B > 0$ such that for any $\theta \in \Theta$, $\|\xi_{\alpha_\theta}\|_{\mathscr{H}} \leq B$. Then with probability at least $1 - \delta$,*

$$\mathbb{P}\big(\mathbf{y} g_{\widehat{\theta}_n}(\mathbf{x}) \leq 0 | \mathscr{D}_n\big) \leq \widehat{\mathscr{R}}_n(\widehat{\theta}_n) + \frac{c_2}{T} + \frac{8BK_\ell}{(1-L)\sqrt{n}} + \frac{2BK_\ell}{1-L}\sqrt{\frac{\log(1/\delta)}{2n}}, \tag{13}$$

*where $c_2 = K_\ell \sup_\theta \left( \|\psi\|_{\text{op}} K_{f_\theta} e^{K_{f_\theta}} \big( L + \|f_\theta\|_\infty e^{K_{f_\theta}} \big) \right)$.*

Close to our result are the bounds obtained by Zhang et al. (2018), Tu et al. (2019), and Chen et al. (2020). The main difference is that the term in $1/T$ does not usually appear, since it comes from the Euler discretization error, whereas the speed in $1/\sqrt{n}$ is the same. For instance, Chen et al. (2020) show that, under some assumptions, the excess risk is of order $\sqrt{de + e^2} T^\alpha K_\ell n^{-1/2}$. We refer to Section 5 for further discussion on the dependency of the different bounds to the parameter $T$. The take-home message is that the detour by continuous-time neural ODE provides a theoretical framework adapted to RNN, at the modest price of an additional $\mathscr{O}(1/T)$ term. Moreover, we note that the bound (13) is 'simple' and holds under mild conditions for a large class of RNN. More precisely, for any recurrent network of the form (1), provided $(A_1)$ and $(A_2)$ are satisfied, then (13) is valid with constants $c_2$ and $B$ depending on the architecture. Such constants are given below in the example of a feedforward RNN. We stress that Theorem 2 can be extended without significant effort to the multi-class classification task, with an appropriate choice of loss function.

**Example 4.** *Take a feedforward RNN with logistic activation, and $\Theta = \{(W, b, \psi) \mid \|W\|_F \leq K_W < (1-L)/32d, \|b\| \leq K_b, \|\psi\|_{\text{op}} \leq K_\psi\}$. Then, Proposition 5 states that $(A_2)$ is satisfied and, with Theorem 1, ensures that*

$$\sup_{\theta \in \Theta} \|\xi_{\alpha_\theta}\|_{\mathscr{H}} \leq \frac{\sqrt{2}K_\psi(1-L)}{1-L-32dK_W} := B, \quad K_{f_\theta} = \max(\|U\|_{\text{op}}, \|V\|_{\text{op}}), \quad \text{and} \quad \|f_\theta\|_\infty = 1.$$

**Sequence-to-sequence learning.** We conclude by showing how to extend both the RKHS embedding of Theorem 1 and the generalization bound of Theorem 2 to the setting of sequence-to-sequence learning. In this case, the output of the network is a sequence

$$g_\theta(\mathbf{x}) = (z_1, \ldots, z_T) \in (\mathbb{R}^p)^T.$$

An immediate extension of Theorem 1 ensures that there exist $p$ elements $\alpha_{1,\theta}, \ldots, \alpha_{p,\theta} \in \mathscr{T}$ such that, for any $1 \leq j \leq T$,

$$\left\| z_j - \left( \langle \alpha_{1,\theta}, S_{[0,j/T]}(\bar{X}) \rangle_{\mathscr{T}}, \ldots, \langle \alpha_{p,\theta}, S_{[0,j/T]}(\bar{X}) \rangle_{\mathscr{T}} \right)^\top \right\| \leq \|\psi\|_{\mathrm{op}} \frac{c_1}{T}. \tag{14}$$

The properties of the signature guarantee that $S_{[0,j/T]}(X) = S(\tilde{X}_{[j]})$ where $\tilde{X}_{[j]}$ is the process equal to $\bar{X}$ on $[0, j/T]$ and then constant on $[j/T, 1]$—see Appendix A.6. With this trick, we have, for any $1 \leq \ell \leq p$, $\langle \alpha_{\ell,\theta}, S_{[0,j/T]}(\bar{X}) \rangle_{\mathscr{T}} = \langle \alpha_{\ell,\theta}, S(\tilde{X}_{[j]}) \rangle_{\mathscr{T}}$, so that we are back in $\mathscr{H}$. Observe that the only difference with (12) is that we consider vector-valued sequential outputs, which requires to introduce the process $\tilde{X}_{[j]}$, but that the rationale is exactly the same.

We let $\ell : (\mathbb{R}^p)^T \times (\mathbb{R}^p)^T \to \mathbb{R}^+$ be the $L_2$ distance, that is, for any $\mathbf{y} = (y_1, \ldots, y_T)$, $\mathbf{y}' = (y'_1, \ldots, y'_T)$, $\ell(\mathbf{y}, \mathbf{y}') = \frac{1}{T} \sum_{j=1}^T \|y_j - y'_j\|^2$. It is assumed that $\mathbf{y}$ takes its values in a compact subset of $\mathbb{R}^q$, i.e., there exists $K_y > 0$ such that $\|y_j\| \leq K_y$.

**Theorem 3.** *Assume that for all $\theta \in \Theta$, $(A_1)$ and $(A_2)$ are verified. Assume, in addition, that there exists a constant $B > 0$ such that for any $1 \leq \ell \leq p$, $\theta \in \Theta$, $\|\xi_{\alpha_{\ell,\theta}}\|_{\mathscr{H}} \leq B$. Then with probability at least $1 - \delta$,*

$$\mathscr{R}(\widehat{\theta}_n) \leq \widehat{\mathscr{R}}_n(\widehat{\theta}_n) + \frac{c_3}{T} + \frac{4pc_4 B(1-L)^{-1}}{\sqrt{n}} + \sqrt{\frac{2c_5 \log(1/\delta)}{n}}, \tag{15}$$

*where $c_3 = \sup_\theta \left( c_{1,\theta} + \|\psi\|_{\mathrm{op}} \|f_\theta\|_\infty \right) + 2\sqrt{p} B(1-L)^{-1} + 2K_y$, $c_4 = B(1-L)^{-1} + K_y$, and $c_5 = 4pB(1-L)^{-1} c_4 + K_y^2$.*

## 3.2 Regularization and stability

In addition to providing a sound theoretical framework, framing deep learning in an RKHS provides a natural norm, which can be used for regularization, as shown for example in the context of convolutional neural networks by Bietti et al. (2019). This regularization ensures stability of predictions, which is crucial in particular in a small sample regime or in the presence of adversarial examples (Gao et al., 2018; Ko et al., 2019). In our binary classification setting, for any inputs $\mathbf{x}, \mathbf{x}' \in (\mathbb{R}^d)^T$, by the Cauchy-Schwartz inequality, we have

$$\|z_T - z'_T\| \leq 2\|\psi\|_{\mathrm{op}} \frac{c_1}{T} + \|\xi_{\alpha_\theta}(\bar{X}) - \xi_{\alpha_\theta}(\bar{X}')\| \leq 2\|\psi\|_{\mathrm{op}} \frac{c_1}{T} + \|\xi_{\alpha_\theta}\|_{\mathscr{H}} \|S(\bar{X}) - S(\bar{X}')\|_{\mathscr{T}}.$$

If $\mathbf{x}$ and $\mathbf{x}'$ are close, so are their associated continuous processes $X$ and $X'$ (which can be approximated for example by taking a piecewise linear interpolation), and so are their signatures. The term $\|S(\bar{X}) - S(\bar{X}')\|_{\mathscr{T}}$ is therefore small (Friz and Victoir, 2010, Proposition 7.66). Therefore, when $T$ is large, we see that the magnitude of $\|\xi_{\alpha_\theta}\|_{\mathscr{H}}$ determines how close the predictions are. A natural training strategy to ensure stable predictions, for the types of networks covered in the present article, is then to penalize the problem by minimizing the loss $\widehat{\mathscr{R}}_n(\theta) + \lambda \|\xi_{\alpha_\theta}\|_{\mathscr{H}}^2$. From a computational point of view, it is possible to compute the norm in $\mathscr{H}$, up to a truncation at $N$ of the Taylor expansion, which we know by Proposition 4 to be reasonable. It remains that computing this norm is a non-trivial task, and implementing smart surrogates is an interesting problem for the future. Note however that computing the signature of the data is not necessary for this regularization strategy.

## 4 Numerical illustrations

This section is here for illustration purposes. Our objective is not to achieve competitive performance, but rather to illustrate the theoretical results. We refer to Appendix D for implementation details.

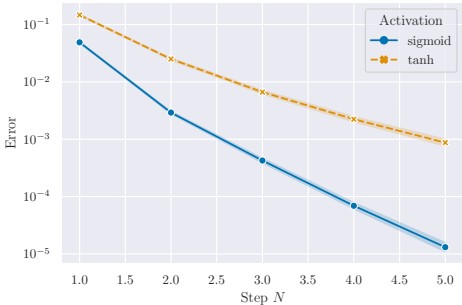
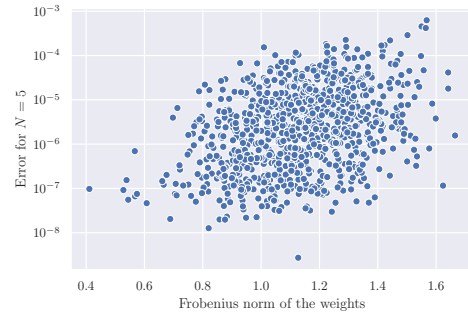

(a) Error on a logarithmic scale as a function of $N$    (b) Error as a function of the norm of the weights

Figure 1: Approximation of the RNN ODE by the step-$N$ Taylor expansion

**Convergence of the Taylor expansion towards the solution of the ODE.**    We illustrate Proposition 4 on a toy example. The process $X$ is a 2-dimensional spiral, and we take feedforward RNN with 2 hidden units. Repeating this procedure with $10^3$ uniform random weight initializations, we observe in Figure 1a that the signature approximation converges exponentially fast in $N$. As seen in Figure 1b, the rate of convergence depends in particular on the norm of the weight matrices, as predicted by Proposition 5. However, condition (11) seems to be over-restrictive, since convergence happens even for weights with norm larger than the bound (we have $1/(8a^2d) \simeq 0.01$ here).

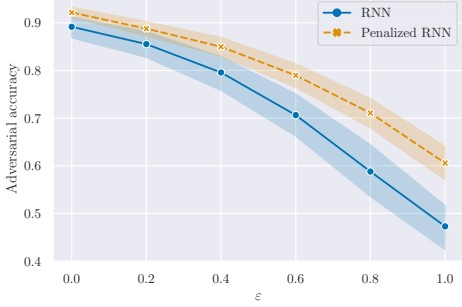

Figure 2: Adversarial accuracy as a function of the adversarial perturbation $\varepsilon$

**Adversarial robustness.**    We illustrate the penalization proposed in Section 3.2 on a toy task that consists in classifying the rotation direction of 2-dimensional spirals. We take a feedforward RNN with 32 hidden units and hyperbolic tangent activation. It is trained on 50 examples, with and without penalization, for 200 epochs. Once trained, the RNN is tested on adversarial examples, generated with the projected gradient descent algorithm with Frobenius norm (Madry et al., 2018), which modifies test examples to maximize the error while staying in a ball of radius $\varepsilon$. We observe in Figure 2 that adding the penalization seems to make the network more stable.

**Comparison of the trained networks.**    The evolution of the Frobenius norm of the weights $\|W\|_F$ and the RKHS norm $\|\xi_{\alpha_\theta}\|_{\mathscr{H}}$ during training is shown in Figure 3. This points out that the penalization, which forces the RNN to keep a small norm in $\mathscr{H}$, leads indeed to learning different weights than the non-penalized RNN. The results also suggest that the Frobenius and RKHS norms are decoupled, since both networks have Frobenius norms of similar magnitude but very different RKHS norms. The figures show one random run, but we observe similar qualitative behavior on others.

## 5 Discussion and conclusion

**Role of the discretization procedure.**    The starting point of the paper was motivated by the fact that the classical residual RNN formulation coincides with an Euler discretization of the ODE (3).

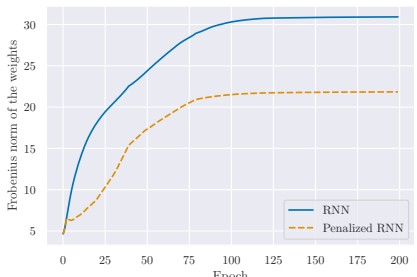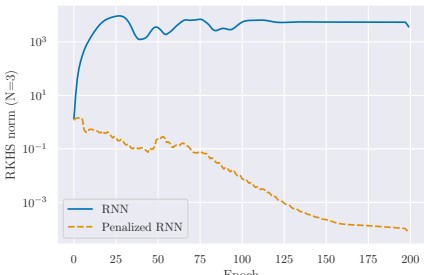

Figure 3: Evolution of the Frobenius norm of the weights and of the RKHS norm during training

This choice of discretization translates into a $1/T$ term in Theorems 2 and 3. However, we could have considered higher-order discretization schemes, such as Runge-Kutta schemes, for which the discretization error decreases as $1/T^p$. Such schemes correspond to alternative architectures, which were already proposed by Wang and Lin (1998), among others. At the limit, we could also consider directly the continuous model (3), as proposed by Chen et al. (2018), in which case the discretization error term vanishes. Of course, such an option requires to be able to sample the continuous-time data at arbitrary times.

**Long-term stability.** RNN are known to be poor at learning long-term dependencies (Bengio et al., 1993; Hochreiter and Schmidhuber, 1997). This is reflected in the literature by performance bounds increasing in $T$, which is not the case of our results (13) and (15), seemingly indicating that we fail to capture this phenomenon. This apparent paradox is related to our assumption that the total variation of $X$ is bounded. Indeed, if a time series is observed for a long time, then its total variation may become large. In this case, it is no longer valid to assume that $\|X\|_{\text{TV}}$ is bounded by $L$. In other words, in our context, the parameter encapsulating the notion of "long-term" is not $T$ but the regularity of $X$ measured by its total variation. Note that the choice of defining $X$ on $[0,1]$ and not another interval $[0,U]$ is arbitrary and does not carry any meaning on the problem of learning long-term dependencies. A thorough analysis of these questions is an interesting research direction for future work.

**Radius of convergence.** The assumptions $\|X\|_{TV;[0,1]} \leq L < 1$ and $\|W\|_F \leq K_W < (1-L)/32d$ can be seen as radii of convergence of the Taylor expansion (10). They allow using the Taylor approximation—which is of a local nature—to prove a global result, the RKHS embedding. In return, the condition on the Frobenius norm of the weights puts restrictions on the admissible parameters of the neural network. However, this bound can be improved, in particular by considering more exotic norms, which we did not explicit for clarity purposes.

**Conclusion.** By bringing together the theory of neural ODE, the signature transform, and kernel methods, we have shown that a recurrent network can be framed in the continuous-time limit as a linear function in a well-chosen RKHS. In addition to giving theoretical insights on the function learned by the network and providing generalization guarantees, this framing suggests regularization strategies to obtain more robust RNN. We have only scratched the surface of the potentialities of leveraging this theory to practical applications, which is a subject of its own and will be tackled in future work.

## Acknowledgements

Authors thank T. Lévy for his inputs on the Picard-Lindelöf theorem and N. Doumèche for fruitful discussion. A. Fermanian has been supported by a grant from Région Île-de-France and P. Marion by a stipend from Corps des Mines.

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
