# Framing RNN as a kernel method: A neural ODE approach
## Supplementary material

## A  Mathematical details

### A.1  Writing the GRU and LSTM in the neural ODE framework

**GRU.**  Recall that the equations of a GRU take the following form: for any $1 \leq j \leq T$,

$$r_{j+1} = \sigma(W_r x_{j+1} + b_r + U_r h_j)$$
$$z_{j+1} = \sigma(W_z x_{j+1} + b_z + U_z h_j)$$
$$n_{j+1} = \tanh\big(W_n x_{j+1} + b_n + r_{j+1} * (U_n h_j + c_n)\big)$$
$$h_{j+1} = (1 - z_{j+1}) * h_j + z_{j+1} * n_{j+1},$$

where $\sigma$ is the logistic activation, tanh the hyperbolic tangent, $*$ the Hadamard product, $r_j$ the reset gate vector, $z_j$ the update gate vector, $W_r, U_r, W_z, U_z, W_n, U_n$ weight matrices, and $b_r, b_z, b_n, c_n$ biases. Since $r_{j+1}$, $z_{j+1}$, and $n_{j+1}$ depend only on $x_{j+1}$ and $h_j$, it is clear that these equations can be rewritten in the form

$$h_{j+1} = h_j + f(h_j, x_{j+1}).$$

We then obtain equation (1) by normalizing $f$ by $1/T$.

**LSTM.**  The LSTM networks are defined, for any $1 \leq j \leq T$, by

$$i_{j+1} = \sigma(W_i x_{j+1} + b_i + U_i h_j)$$
$$f_{j+1} = \sigma(W_f x_{j+1} + b_f + U_f h_j)$$
$$g_{j+1} = \tanh(W_g x_{j+1} + b_g + U_g h_j)$$
$$o_{j+1} = \sigma(W_o x_{j+1} + b_o + U_o h_j)$$
$$c_{j+1} = f_{j+1} * c_j + i_{j+1} * g_{j+1}$$
$$h_{j+1} = o_{j+1} * \tanh(c_{j+1}),$$

where $\sigma$ is the logistic activation, tanh the hyperbolic tangent, $*$ the Hadamard product, $i_j$ the input gate, $f_j$ the forget gate, $g_j$ the cell gate, $o_j$ the output gate, $c_j$ the cell state, $W_i, U_i, W_f, U_f, W_g, U_g$ $W_o, U_o$ weight matrices, and $b_i, b_f, b_g, b_o$ biases. Since $i_{j+1}, f_{j+1}, g_{j+1}, o_{j+1}$ depend only on $x_{j+1}$ and $h_j$, these equations can be rewritten in the form

$$h_{j+1} = f_1(h_j, x_{j+1}, c_{j+1})$$
$$c_{j+1} = f_2(h_j, x_{j+1}, c_j).$$

Let $\tilde{h}_j = (h_j^\top, c_j^\top)^\top$ be the hidden state defined by stacking the hidden and cell state. Then, clearly, $\tilde{h}$ follows an equation of the form

$$\tilde{h}_{j+1} = f(\tilde{h}_j, x_{j+1}).$$

We obtain (1) by subtracting $\tilde{h}_j$ and normalizing by $1/T$.

### A.2  Picard-Lindelöf theorem

Consider a CDE of the form (8). We recall the Picard-Lindelöf theorem as given by Lyons et al. (2007, Theorem 1.3), and provide a proof for the sake of completeness.

**Theorem 4** (Picard-Lindelöf theorem). *Assume that $X \in BV^c([0,1], \mathbb{R}^d)$ and that $\mathbf{F}$ is Lipschitz-continuous with constant $K_{\mathbf{F}}$. Then, for any $H_0 \in \mathbb{R}^e$, the differential equation (8) admits a unique solution $H : [0,1] \to \mathbb{R}^e$.*

*Proof.* Let $\mathscr{C}([s,t]), \mathbb{R}^e)$ be the set of continuous functions from $[s,t]$ to $\mathbb{R}^e$. For any $[s,t] \subset [0,1]$, $\zeta \in \mathbb{R}^e$, let $\Psi$ be the function

$$\Psi : \mathscr{C}([s,t]), \mathbb{R}^e) \to \mathscr{C}([s,t], \mathbb{R}^e)$$

$$Y \mapsto \big(v \mapsto \zeta + \int_s^v \mathbf{F}(Y_u) dX_u\big).$$

For any $Y, Y' \in \mathscr{C}([s,t]), \mathbb{R}^e), v \in [s,t]$,

$$\|\Psi(Y)_v - \Psi(Y')_v\| \leq \int_s^v \left\| \big(\mathbf{F}(Y_u) - \mathbf{F}(Y'_u)\big)dX_u \right\|$$

$$\leq \int_s^v \|\mathbf{F}(Y_u) - \mathbf{F}(Y'_u)\|_{\mathrm{op}}\|dX_u\|$$

$$\leq \int_s^v K_{\mathbf{F}}\|Y_u - Y'_u\|\|dX_u\|$$

$$\leq K_{\mathbf{F}}\|Y - Y'\|_\infty \int_s^v \|dX_u\|$$

$$\leq K_{\mathbf{F}}\|Y - Y'\|_\infty \|X\|_{TV;[s,t]}.$$

This shows that the function $\Psi$ is Lipschitz-continuous on $\mathscr{C}([s,t]), \mathbb{R}^e)$ endowed with the supremum norm, with Lipschitz constant $K_{\mathbf{F}}\|X\|_{TV;[s,t]}$. Clearly, the function $t \mapsto \|X\|_{TV;[0,t]}$ is non-decreasing and uniformly continuous on the compact interval $[0,1]$. Therefore, for any $\varepsilon > 0$, there exists $\delta > 0$ such that

$$|t - s| < \delta \Rightarrow \big| \|X\|_{TV;[0,t]} - \|X\|_{TV;[0,s]} \big| < \varepsilon.$$

Take $\varepsilon = 1/K_{\mathbf{F}}$. Then on any interval $[s,t]$ of length smaller than $\delta$, one has $\|X\|_{TV;[s,t]} = \|X\|_{TV;[0,t]} - \|X\|_{TV;[0,s]} < 1/K_{\mathbf{F}}$, so that the function $\Psi$ is a contraction. By the Banach fixed-point theorem, for any initial value $\zeta$, $\Psi$ has a unique fixed point. Hence, there exists a solution to (8) on any interval of length $\delta$ with any initial condition. To obtain a solution on $[0,1]$ it is sufficient to concatenate these solutions. □

A corollary of this theorem is a Picard-Lindelöf theorem for initial value problems of the form

$$dH_t = f(H_t, X_t)dt, \quad H_0 = \zeta, \tag{16}$$

where $f : \mathbb{R}^e \times \mathbb{R}^d \to \mathbb{R}^e, \zeta \in \mathbb{R}^e$.

**Corollary 1.** *Assume that $f$ is Lipschitz continuous in its first variable. Then, for any $\zeta \in \mathbb{R}^e$, the initial value problem (16) admits a unique solution.*

*Proof.* Let $f_X : (h,t) \mapsto f(h, X_t)$. Then the solution of (16) is solution of the differential equation

$$dH_t = f_X(H_t, t)dt.$$

Let $d = 1, \bar{e} = e + 1$, and $\mathbf{F}$ be the vector field defined by

$$\mathbf{F} : h \mapsto \begin{pmatrix} f_X(h^{1:e}, h^{e+1}) \\ 1 \end{pmatrix},$$

where $h^{1:e}$ denotes the projection of $h$ on its first $e$ coordinates. Then, since $f_X$ is Lipschitz, so is the vector field $\mathbf{F}$. Theorem 4 therefore applies to the differential equation

$$dH_t = \mathbf{F}(H_t)dt, \quad H_0 = (\zeta^\top, 0)^\top.$$

Projecting this differential equation on the last coordinate gives $dH_t^{e+1} = dt$, that is, $H_t^{e+1} = t$. Projecting on the first $e$ coordinates exactly provides equation (16), which therefore has a unique solution, equal to $H^{1:e}$. □

### A.3 Operator norm

**Definition 3.** *Let $(E, \|\cdot\|_E)$ and $(F, \|\cdot\|_F)$ be two normed vector spaces and let $f \in \mathscr{L}(E, F)$, where $\mathscr{L}(E, F)$ is the space of linear functions from $E$ to $F$. The operator norm of $f$ is defined by*

$$\|f\|_{\mathrm{op}} = \sup_{u \in E, \|u\|_E = 1} \|f(u)\|_F.$$

*Equipped with this norm, $\mathscr{L}(E, F)$ is a normed vector space.*

This definition is valid when $f$ is represented by a matrix.

### A.4 Tensor Hilbert space

Let us first briefly recall some elements on tensor spaces. If $e_1, \ldots, e_d$ is the canonical basis of $\mathbb{R}^d$, then $(e_{i_1} \otimes \cdots \otimes e_{i_k})_{1 \leq i_1, \ldots, i_k \leq d}$ is a basis of $(\mathbb{R}^d)^{\otimes k}$. Any element $a \in (\mathbb{R}^d)^{\otimes k}$ can therefore be written as

$$a = \sum_{1 \leq i_1, \ldots, i_k \leq d} a^{(i_1, \ldots, i_k)} e_{i_1} \otimes \cdots \otimes e_{i_k},$$

where $a^{(i_1, \ldots, i_k)} \in \mathbb{R}$. The tensor space $(\mathbb{R}^d)^{\otimes k}$ is a Hilbert space of dimension $d^k$, with scalar product

$$\langle a, b \rangle_{(\mathbb{R}^d)^{\otimes k}} = \sum_{1 \leq i_1, \ldots, i_k \leq d} a^{(i_1, \ldots, i_k)} b^{(i_1, \ldots, i_k)}$$

and associated norm $\| \cdot \|_{(\mathbb{R}^d)^{\otimes k}}$.

We now consider the space $\mathscr{T}$ defined by (6). The sum, multiplication by a scalar, and scalar product on $\mathscr{T}$ are defined as follows: for any $a = (a_0, \ldots, a_k, \ldots) \in \mathscr{T}$, $b = (b_0, \ldots, b_k, \ldots) \in \mathscr{T}$, $\lambda \in \mathbb{R}$,

$$a + \lambda b = (a_0 + \lambda b_0, \ldots, a_k + \lambda b_k, \ldots) \quad \text{and} \quad \langle a, b \rangle_{\mathscr{T}} = \sum_{k=0}^{\infty} \langle a_k, b_k \rangle_{(\mathbb{R}^d)^{\otimes k}},$$

with the convention $(\mathbb{R}^d)^{\otimes 0} = \mathbb{R}$.

**Proposition 6.** $(\mathscr{T}, +, \cdot, \langle \cdot, \cdot \rangle_{\mathscr{T}})$ *is a Hilbert space.*

*Proof.* By the Cauchy-Schwartz inequality, $\langle \cdot, \cdot \rangle_{\mathscr{T}}$ is well-defined: for any $a, b \in \mathscr{T}$,

$$|\langle a, b \rangle_{\mathscr{T}}| \leq \sum_{k=0}^{\infty} |\langle a_k, b_k \rangle_{(\mathbb{R}^d)^{\otimes k}}| \leq \sum_{k=0}^{\infty} \|a_k\|_{(\mathbb{R}^d)^{\otimes k}} \|b_k\|_{(\mathbb{R}^d)^{\otimes k}}$$

$$\leq \Big( \sum_{k=0}^{\infty} \|a_k\|_{(\mathbb{R}^d)^{\otimes k}}^2 \Big)^{1/2} \Big( \sum_{k=0}^{\infty} \|b_k\|_{(\mathbb{R}^d)^{\otimes k}}^2 \Big)^{1/2} < \infty.$$

Moreover, $\mathscr{T}$ is a vector space: for any $a, b \in \mathscr{T}$, $\lambda \in \mathbb{R}$, since

$$a + \lambda b = (a_0 + \lambda b_0, \ldots, a_k + \lambda b_k, \ldots),$$

and

$$\sum_{k=0}^{\infty} \|a_k + \lambda b_k\|_{(\mathbb{R}^d)^{\otimes k}}^2 = \sum_{k=0}^{\infty} \|a_k\|_{(\mathbb{R}^d)^{\otimes k}}^2 + \lambda^2 \sum_{k=0}^{\infty} \|b_k\|_{(\mathbb{R}^d)^{\otimes k}}^2$$

$$+ 2\lambda \sum_{k=0}^{\infty} \langle a_k, b_k \rangle_{(\mathbb{R}^d)^{\otimes k}}$$

$$\leq \sum_{k=0}^{\infty} \|a_k\|_{(\mathbb{R}^d)^{\otimes k}}^2 + \lambda^2 \sum_{k=0}^{\infty} \|b_k\|_{(\mathbb{R}^d)^{\otimes k}}^2 + 2\lambda \langle a, b \rangle_{\mathscr{T}} < \infty,$$

we see that $a + \lambda b \in \mathscr{T}$. The operation $\langle \cdot, \cdot \rangle_{\mathscr{T}}$ is also bilinear, symmetric, and positive definite:

$$\langle a, a \rangle_{\mathscr{T}} = 0 \Leftrightarrow \sum_{k=0}^{\infty} \|a_k\|_{(\mathbb{R}^d)^{\otimes k}}^2 = 0 \Leftrightarrow \forall k \in \mathbb{N}, \|a_k\|_{(\mathbb{R}^d)^{\otimes k}}^2 = 0 \Leftrightarrow \forall k \in \mathbb{N}, a_k = 0 \Leftrightarrow a = 0.$$

Therefore $\langle \cdot, \cdot \rangle_{\mathscr{T}}$ is an inner product on $\mathscr{T}$. Finally, let $(a^{(n)})_{n \in \mathbb{N}}$ be a Cauchy sequence in $\mathscr{T}$. Then, for any $n, m \geq 0$,

$$\|a^{(n)} - a^{(m)}\|_{\mathscr{T}}^2 = \sum_{k=0}^{\infty} \|a_k^{(n)} - a_k^{(m)}\|_{(\mathbb{R}^d)^{\otimes k}}^2,$$

so for any $k \in \mathbb{N}$, the sequence $(a_k^{(n)})_{n \in \mathbb{N}}$ is Cauchy in $(\mathbb{R}^d)^{\otimes k}$. Since $(\mathbb{R}^d)^{\otimes k}$ is a Hilbert space, $(a_k^{(n)})_{n \in \mathbb{N}}$ converges to a limit $a_k^{(\infty)} \in (\mathbb{R}^d)^{\otimes k}$. Let $a^{(\infty)} = (a_0^{(\infty)}, \ldots, a_k^{(\infty)}, \ldots)$. To finish the

proof, we need to show that $a^{(\infty)} \in \mathscr{T}$ and that $a^{(n)}$ converges to $a^{(\infty)}$ in $\mathscr{T}$. First, note that there exists a constant $B > 0$ such that for any $n \in \mathbb{N}$,

$$\|a^{(n)}\|_{\mathscr{T}} \leq B.$$

To see this, observe that for $\varepsilon > 0$, there exists $N \in \mathbb{N}$ such that for any $n \geq N$, $\|a^{(n)} - a^{(N)}\|_{\mathscr{T}} < \varepsilon$, and so $\|a^{(n)}\|_{\mathscr{T}} \leq \varepsilon + \|a^{(N)}\|_{\mathscr{T}}$. Take $B = \max(\|a^{(1)}\|_{\mathscr{T}}, \ldots, \|a^{(N)}\|_{\mathscr{T}}, \varepsilon + \|a^{(N)}\|_{\mathscr{T}})$. Then, for any $K \in \mathbb{N}$,

$$\sum_{k=0}^{K} \|a_k^{(n)}\|_{(\mathbb{R}^d)^{\otimes k}}^2 \leq \|a^{(n)}\|_{\mathscr{T}} \leq B.$$

Letting $K \to \infty$, we obtain that $\|a^{(\infty)}\|_{\mathscr{T}} \leq B$, and therefore $a^{(\infty)} \in \mathscr{T}$. Finally, let $\varepsilon > 0$ and let $N \in \mathbb{N}$ be such that for any $n, m \geq N$, $\|a^{(n)} - a^{(m)}\|_{\mathscr{T}} < \varepsilon$. Clearly, for any $K \in \mathbb{N}$,

$$\sum_{k=0}^{K} \|a_k^{(n)} - a_k^{(m)}\|_{(\mathbb{R}^d)^{\otimes k}}^2 < \varepsilon^2.$$

Letting $m \to \infty$ leads to

$$\sum_{k=1}^{K} \|a_k^{(n)} - a_k^{(\infty)}\|_{(\mathbb{R}^d)^{\otimes k}}^2 < \varepsilon^2,$$

and letting $K \to \infty$ gives

$$\|a^{(n)} - a^{(\infty)}\|_{\mathscr{T}} < \varepsilon,$$

which completes the proof. □

### A.5 Bounding the derivatives of the logistic and hyperbolic tangent activations

**Lemma 1.** *Let $\sigma$ be the logistic function defined, for any $x \in \mathbb{R}$, by $\sigma(x) = 1/(1+e^{-x})$. Then, for any $n \geq 0$,*

$$\|\sigma^{(n)}\|_\infty \leq 2^{n-1} n!.$$

*Proof.* For any $x \in \mathbb{R}$, one has (Minai and Williams, 1993, Theorem 2)

$$\sigma^{(n)}(x) = \sum_{k=1}^{n+1} (-1)^{k-1}(k-1)! \left\{ {n+1 \atop k} \right\} \sigma(x)^k,$$

where $\left\{ {n \atop k} \right\}$ stands for the Stirling number of the second kind (see, e.g., Riordan, 1958). Let

$$u_n = \sum_{k=1}^{n+1} (k-1)! \left\{ {n+1 \atop k} \right\}$$

for $n \geq 1$ and $u_0 = 1$. Since $0 \leq \sigma(x) \leq 1$, it is clear that $|\sigma^{(n)}(x)| \leq u_n$. Using the fact that the Stirling numbers satisfy the recurrence relation

$$\left\{ {n+1 \atop k} \right\} = k \left\{ {n \atop k} \right\} + \left\{ {n \atop k-1} \right\},$$

valid for all $0 \leq k \leq n$, we have

$$u_n = \sum_{k=1}^{n} (k-1)! \left( k \left\{ {n \atop k} \right\} + \left\{ {n \atop k-1} \right\} \right) + n! = \sum_{k=1}^{n} k! \left\{ {n \atop k} \right\} + \sum_{k=0}^{n-1} k! \left\{ {n \atop k} \right\} + n! = 2 \sum_{k=1}^{n} k! \left\{ {n \atop k} \right\}$$

$$(\text{since } \left\{ {n \atop 0} \right\} = 0)$$

$$\leq 2n \sum_{k=1}^{n} (k-1)! \left\{ {n \atop k} \right\} = 2n u_{n-1}.$$

Thus, by induction, $u_n \leq 2^{n-1} n!$, from which the claim follows. □

**Lemma 2.** *Let* tanh *be the hyperbolic tangent function. Then, for any $n \geq 0$,*

$$\|\tanh^{(n)}\|_{\infty} \leq 4^n n! \,.$$

*Proof.* Let $\sigma$ be the logistic function. Straightforward calculations yield the equality, valid for any $x \in \mathbb{R}$,

$$\tanh(x) = 2\sigma(2x) - 1.$$

But, for any $n \geq 1$,

$$\tanh^{(n)}(x) = 2^{n+1}\sigma^{(n)}(2x),$$

and thus, by Lemma 1,

$$\|\tanh^{(n)}\|_{\infty} \leq 2^{n+1}\|\sigma^{(n)}\|_{\infty} \leq 4^n n! \,.$$

The inequality is also true for $n = 0$ since $\|\tanh\|_{\infty} \leq 1$. $\qquad\square$

### A.6 Chen's formula

First, note that it is straightforward to extend the definition of the signature to any interval $[s, t] \subset [0, 1]$. The next proposition, known as Chen's formula (Lyons et al., 2007, Theorem 2.9), tells us that the signature can be computed iteratively as tensor products of signatures on subintervals.

**Proposition 7.** *Let $X \in BV^c([s, t], \mathbb{R}^d)$ and $u \in (s, t)$. Then*

$$S_{[s,t]}(X) = S_{[s,u]}(X) \otimes S_{[u,t]}(X).$$

Next, it is clear that the signature of a constant path is equal to $\mathbf{1} = (1, 0, \ldots, 0, \ldots)$ which is the null element in $\mathscr{T}$. Indeed, let $Y \in BV^c([s, t], \mathbb{R}^d)$ be a constant path. Then, for any $k \geq 1$,

$$\mathbb{Y}_{[s,t]}^k = k! \int \cdots \int_{s \leq u_1 < \cdots < u_k \leq t} dY_{u_1} \otimes \cdots \otimes dY_{u_k} = k! \int \cdots \int_{s \leq u_1 < \cdots < u_k \leq t} 0 \otimes \cdots \otimes 0 = 0.$$

Now let $X \in BV^c([0, 1], \mathbb{R}^d)$ and consider the path $\tilde{X}_{[j]}$ equal to the time-augmented path $\bar{X}$ on $[0, j/T]$ and then constant on $[j/T, 1]$—see Figure 4. We have by Proposition 7

$$S_{[0,1]}(\tilde{X}_{[j]}) = S_{[0,j/T]}(\tilde{X}_{[j]}) \otimes S_{[j/T,1]}(\tilde{X}_{[j]}) = S_{[0,j/T]}(\bar{X}) \otimes \mathbf{1} = S_{[0,j/T]}(\bar{X}).$$

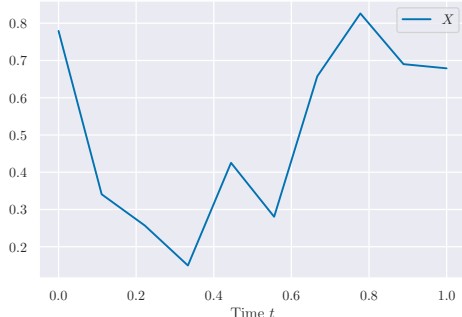 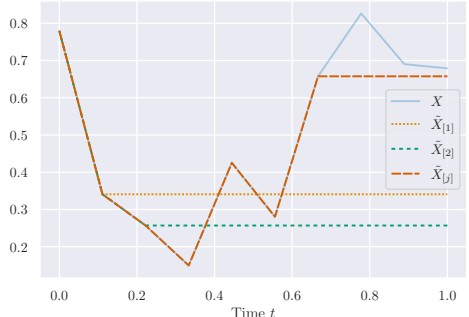

Figure 4: Example of a path $X \in BV^c([0, 1], \mathbb{R})$ (left) and its corresponding paths $\tilde{X}_{[j]}$, plotted against time, for different values of $j \in \{1, \ldots, T\}$ (right)

## B Proofs

### B.1 Proof of Proposition 1

According to Assumption $(A_1)$, for any $h_1, h_2 \in \mathbb{R}^e$, $x_1, x_2 \in \mathbb{R}^d$, one has

$$\|f(h_1, x_1) - f(h_2, x_1)\| \leq K_f \|h_1 - h_2\| \quad \text{and} \quad \|f(h_1, x_1) - f(h_1, x_2)\| \leq K_f \|x_1 - x_2\|.$$

Under assumption $(A_1)$, by Corollary 1, the initial value problem (3) admits a unique solution $H$. Let us first show that for any $t \in [0, 1]$, $H_t$ is bounded independently of $X$. For any $t \in [0, 1]$,

$$
\begin{aligned}
\|H_t - H_0\| = \Big\| \int_0^t f(H_u, X_u) du \Big\| &\leq \int_0^t \|f(H_u, X_u)\| du \\
&= \int_0^t \|f(H_u, X_u) - f(H_0, X_u) + f(H_0, X_u)\| du \\
&\leq \int_0^t \|f(H_u, X_u) - f(H_0, X_u)\| + \int_0^t \|f(H_0, X_u)\| du \\
&\leq K_f \int_0^t \|H_u - H_0\| du + t \sup_{\|x\| \leq L} \|f(H_0, x)\|.
\end{aligned}
$$

Applying Grönwall's inequality to the function $t \mapsto \|H_t - H_0\|$ yields

$$
\|H_t - H_0\| \leq t \sup_{\|x\| \leq L} \|f(H_0, x)\| \exp \Big( \int_0^t K_f du \Big) \leq \sup_{\|x\| \leq L} \|f(H_0, x)\| e^{K_f} := M.
$$

Given that $H_0 = h_0 = 0$, we conclude that $\|H_t\| \leq M$.

Next, let

$$
\|f\|_\infty = \sup_{\|x\| \leq L, \|h\| \leq M} f(h, x).
$$

By similar arguments, for any $[s, t] \subset [0, 1]$, Grönwall's inequality applied to the function $t \mapsto \|H_t - H_s\|$ yields

$$
\|H_t - H_s\| \leq (t - s) \|f\|_\infty e^{K_f}.
$$

Therefore, for any partition $(t_0, \ldots, t_k)$ of $[s, t]$,

$$
\sum_{i=1}^k \|H_{t_i} - H_{t_{i-1}}\| \leq \|f\|_\infty e^{K_f} \sum_{i=1}^k (t_i - t_{i-1}) \leq \|f\|_\infty e^{K_f} (t - s),
$$

and, taking the supremum over all partitions of $[s, t]$, $\|H\|_{TV;[s,t]} \leq \|f\|_\infty e^{K_f} (t - s)$. In other words, $H$ is of bounded variation on any interval $[s, t] \subset [0, 1]$. Let $(t_0, \ldots, t_T)$ denote the regular partition of $[0, 1]$ with $t_j = j/T$. For any $1 \leq j \leq T$, we have

$$
\begin{aligned}
\|H_{t_j} - h_j\| = \Big\| H_{t_{j-1}} + \int_{t_{j-1}}^{t_j} f(H_u, X_u) du - h_{j-1} - \frac{1}{T} f(h_{j-1}, x_j) \Big\| \\
\leq \|H_{t_{j-1}} - h_{j-1}\| + \int_{t_{j-1}}^{t_j} \big\| f(H_u, X_u) - f(h_{j-1}, x_j) \big\| du.
\end{aligned}
$$

Writing

$$
\begin{aligned}
\big\| f(H_u, X_u) - f(h_{j-1}, x_j) \big\| = \big\| f(H_u, X_u) - f(H_u, x_j) + f(H_u, x_j) - f(h_{j-1}, x_j) \big\| \\
\leq \big\| f(H_u, X_u) - f(H_u, x_j) \big\| + \big\| f(H_u, x_j) - f(h_{j-1}, x_j) \big\| \\
\leq K_f \big\| X_u - x_j \big\| + K_f \big\| H_u - h_{j-1} \big\|,
\end{aligned}
$$

we obtain

$$
\begin{aligned}
\|H_{t_j} - h_j\| &\leq \|H_{t_{j-1}} - h_{j-1}\| + K_f \int_{t_{j-1}}^{t_j} \|H_u - h_{j-1}\| du + K_f \int_{t_{j-1}}^{t_j} \|X_u - x_j\| du \\
&\leq \|H_{t_{j-1}} - h_{j-1}\| + K_f \int_{t_{j-1}}^{t_j} \big( \|H_u - H_{t_{j-1}}\| + \|H_{t_{j-1}} - h_{j-1}\| \big) du \\
&\quad + \frac{K_f}{T} \|X\|_{TV;[t_{j-1}, t_j]} \\
&\leq \Big( 1 + \frac{K_f}{T} \Big) \|H_{t_{j-1}} - h_{j-1}\| + \frac{K_f}{T} \big( \|H\|_{TV;[t_{j-1}, t_j]} + \|X\|_{TV;[t_{j-1}, t_j]} \big).
\end{aligned}
$$

By induction, we are led to

$$\|H_{t_j} - h_j\| \le \frac{K_f}{T} \sum_{k=0}^{j-1} \left(1 + \frac{K_f}{T}\right)^k \left(\|H\|_{TV;[t_k,t_{k+1}]} + \|X\|_{TV;[t_k,t_{k+1}]}\right)$$

$$\le \frac{K_f}{T} \left(1 + \frac{K_f}{T}\right)^T \left(\|X\|_{TV;[0,1]} + \|H\|_{TV;[0,1]}\right)$$

$$\le \frac{K_f e^{K_f}}{T} \left(L + \|f\|_\infty e^{K_f}\right),$$

which concludes the proof.

## B.2 Proof of Proposition 2

Let $\bar{h} \in \mathbb{R}^{\bar{e}}$ and let $\bar{h}^{i:j} = (\bar{h}^i, \ldots, \bar{h}^j)$ be its projection on a subset of coordinates. It is sufficient to take $\mathbf{F}$ defined by

$$\mathbf{F}(\bar{h}) = \begin{pmatrix} 0_{e \times d} & \frac{2}{1-L} f(\bar{h}^{1:e}, \bar{h}^{e+1:e+d}) \\ I_{d \times d} & 0_{d \times 1} \end{pmatrix},$$

where $I_{d \times d}$ denotes the identity matrix and $0_{\cdot \times \cdot}$ the matrix full of zeros. The function $\bar{H}$ is then solution of

$$d\bar{H}_t = \begin{pmatrix} 0_{e \times d} & \frac{2}{1-L} f(\bar{H}_t^{1:e}, \bar{H}_t^{e+1:e+d}) \\ I_{d \times d} & 0_{d \times 1} \end{pmatrix} \begin{pmatrix} dX_t \\ \frac{1-L}{2} dt \end{pmatrix}.$$

Note that under assumption $(A_1)$, the tensor field $\mathbf{F}$ satisfies the assumptions of the Picard-Lindelöf theorem (Theorem 4) so that $\bar{H}$ is well-defined. The projection of this equation on the last $d$ coordinates gives

$$d\bar{H}_t^{e+1:e+d} = dX_t, \quad \bar{H}_0^{e+1:e+d} = X_0,$$

and therefore $\bar{H}_t^{e+1:e+d} = X_t$. The projection on the first $e$ coordinates gives

$$d\bar{H}_t^{1:e} = \frac{2}{1-L} f(\bar{H}_t^{1:e}, X_t) \frac{1-L}{2} dt = f(\bar{H}_t^{1:e}, X_t) dt, \quad \bar{H}_0^{1:e} = h_0,$$

which is exactly (3).

## B.3 Proof of Proposition 3

According to Lyons (2014, Lemma 5.1), one has

$$\|\bar{\mathbb{X}}_{[0,t]}^k\|_{(\mathbb{R}^d)^{\otimes k}} \le \|\bar{X}\|_{TV;[0,t]}^k.$$

Let $(t_0, \ldots, t_k)$ be a partition of $[0, t]$. Then

$$\sum_{j=1}^{k} \|\bar{X}_{t_j} - \bar{X}_{t_{j-1}}\| = \sum_{j=1}^{k} \sqrt{\|X_{t_j} - X_{t_{j-1}}\|^2 + \left(\frac{1-L}{2}\right)^2 (t_j - t_{j-1})^2}$$

$$\le \sum_{j=1}^{k} \|X_{t_j} - X_{t_{j-1}}\| + \frac{1-L}{2} \sum_{j=1}^{k} (t_j - t_{j-1})$$

$$= \sum_{j=1}^{k} \|X_{t_j} - X_{t_{j-1}}\| + \frac{1-L}{2} t.$$

Taking the supremum over any partition of $[0, t]$ we obtain

$$\|\bar{X}\|_{TV;[0,t]} \le \|X\|_{TV;[0,t]} + \frac{1-L}{2} t \le L + \frac{1-L}{2} = \frac{1+L}{2} < 1,$$

and thus $\|\bar{\mathbb{X}}_{[0,t]}^k\|_{(\mathbb{R}^d)^{\otimes k}} \le \left(\frac{1+L}{2}\right)^k$. It is then clear that

$$\|S_{[0,t]}(\bar{X})\|_{\mathscr{T}} = \left(\sum_{k=0}^{\infty} \|\bar{\mathbb{X}}_{[0,t]}^k\|_{(\mathbb{R}^d)^{\otimes k}}^2\right)^{1/2} \le \sum_{k=0}^{\infty} \|\bar{\mathbb{X}}_{[0,t]}^k\|_{(\mathbb{R}^d)^{\otimes k}} \le \sum_{k=0}^{\infty} \left(\frac{1+L}{2}\right)^k = 2(1-L)^{-1}.$$

## B.4 Proof of Proposition 4

We first recall the fundamental theorem of calculus for line integrals (also known as gradient theorem).

**Theorem 5.** *Let $g : \mathbb{R}^e \to \mathbb{R}$ be a continuously differentiable function, and let $\gamma : [a, b] \to \mathbb{R}^e$ be a smooth curve in $\mathbb{R}^e$. Then*

$$\int_a^b \nabla g(\gamma_t) d\gamma_t = g(\gamma_b) - g(\gamma_a),$$

*where $\nabla g$ denotes the gradient of $g$.*

The identity above immediately generalizes to a function $g : \mathbb{R}^e \to \mathbb{R}^e$:

$$\int_a^b J(g)(\gamma_t) d\gamma_t = g(\gamma_b) - g(\gamma_a),$$

where $J(g) \in \mathbb{R}^{e \times e}$ is the Jacobian matrix of $g$. Let us apply Theorem 5 to the vector field $F^i$ between 0 and $t$, with $\gamma = H$. We have

$$F^i(H_t) - F^i(H_0) = \int_0^t J(F^i)(H_u) dH_u = \int_0^t J(F^i)(H_u) \sum_{j=1}^d F^j(H_u) dX_u$$

$$= \sum_{j=1}^d \int_0^t J(F^i)(H_u) F^j(H_u) dX_u = \sum_{j=1}^d \int_0^t F^j \star F^i(H_u) dX_u.$$

Iterating this procedure $(N-1)$ times for the vector fields $F^1, \ldots, F^d$ yields

$$H_t = H_0 + \sum_{i=1}^d \int_0^t F^i(H_u) dX_u^i$$

$$= H_0 + \sum_{i=1}^d \int_0^t F^i(H_0) dX_u^i + \sum_{i=1}^d \int_0^t \sum_{j=1}^d \int_0^u F^j \star F^i(H_v) dX_v^j dX_u^i$$

$$= H_0 + \sum_{i=1}^d F^i(H_0) S_{[0,t]}^{(i)}(X) + \sum_{1 \leq i,j \leq d} \int_{0 \leq v \leq u \leq t} F^j \star F^i(H_v) dX_v^j dX_u^i$$

$$= \cdots$$

$$= H_0 + \sum_{k=1}^N \sum_{1 \leq i_1,\ldots,i_k \leq d} F^{i_1} \star \cdots \star F^{i_k}(H_0) \frac{1}{k!} S_{[0,t]}^{(i_1,\ldots,i_k)}(X)$$

$$+ \sum_{1 \leq i_1,\ldots,i_{N+1} \leq d} \int_{\Delta_{N+1;[0,t]}} F^{i_1} \star \cdots \star F^{i_{N+1}}(H_{u_1}) dX_{u_1}^{i_1} \cdots dX_{u_{N+1}}^{i_{N+1}},$$

where $\Delta_{N;[0,t]} := \{(u_1, \cdots, u_N) \in [0, t]^N \,|\, 0 \leq u_1 < \cdots < u_N \leq t\}$ is the simplex in $[0, t]^N$. The first $(N+1)$ terms equal $H_t^N$. Hence,

$$\|H_t - H_t^N\|$$

$$= \left\| \sum_{1 \leq i_1,\ldots,i_{N+1} \leq d} \int_{\Delta_{N+1;[0,t]}} F^{i_1} \star \cdots \star F^{i_{N+1}}(H_{u_1}) dX_{u_1}^{i_1} \cdots dX_{u_{N+1}}^{i_{N+1}} \right\|$$

$$\leq \sum_{1 \leq i_1,\ldots,i_{N+1} \leq d} \int_{\Delta_{N+1;[0,t]}} \|F^{i_1} \star \cdots \star F^{i_{N+1}}(H_{u_1})\| |dX_{u_1}^{i_1}| \cdots |dX_{u_{N+1}}^{i_{N+1}}|$$

$$\leq \sum_{1 \leq i_1,\ldots,i_{N+1} \leq d} \int_{\Delta_{N+1;[0,t]}} \sup_{1 \leq i_1,\ldots,i_{N+1} \leq d, \|h\| \leq M} \|F^{i_1} \star \cdots \star F^{i_{N+1}}(h)\| |dX_{u_1}^{i_1}| \cdots |dX_{u_{N+1}}^{i_{N+1}}|$$

$$\leq \Lambda_{N+1}(\mathbf{F}) \sum_{1 \leq i_1,\ldots,i_{N+1} \leq d} \int_{\Delta_{N+1;[0,t]}} |dX_{u_1}^{i_1}| \cdots |dX_{u_{N+1}}^{i_{N+1}}|.$$

Thus,

$$
\begin{aligned}
\|H_t - H_t^N\| &\leq \Lambda_{N+1}(\mathbf{F}) \sum_{1 \leq i_1, \ldots, i_{N+1} \leq d} \int_{\Delta_{N+1;[0,t]}} |dX_{u_1}^{i_1}| \cdots |dX_{u_{N+1}}^{i_{N+1}}| \\
&\leq \Lambda_{N+1}(\mathbf{F}) \sum_{1 \leq i_1, \ldots, i_{N+1} \leq d} \int_{\Delta_{N+1;[0,t]}} \|dX_{u_1}\| \cdots \|dX_{u_{N+1}}\| \\
&= \Lambda_{N+1}(\mathbf{F}) \frac{d^{N+1}}{(N+1)!} \int_{[0,t]^{N+1}} \|dX_{u_1}\| \cdots \|dX_{u_{N+1}}\| \\
&= \Lambda_{N+1}(\mathbf{F}) \frac{d^{N+1}}{(N+1)!} \Big( \int_0^t \|dX_u\| \Big)^{N+1} \\
&= \Lambda_{N+1}(\mathbf{F}) \frac{d^{N+1}}{(N+1)!} \|X\|_{TV;[0,t]}^{N+1} \leq \Lambda_{N+1}(\mathbf{F}) \frac{d^{N+1}}{(N+1)!}.
\end{aligned}
$$

## B.5 Proof of Proposition 5

For simplicity of notation, since the context is clear, we now use the notation $\|\cdot\|$ instead of $\|\cdot\|_{(\mathbb{R}^e)^{\otimes k}}$. According to Proposition 1, the solution $\bar{H}$ of (4) verifies $\|\bar{H}_t\| \leq M + L := \bar{M}$. We therefore place ourselves in the ball $\mathscr{B}_{\bar{M}}$. Recall that for any $1 \leq i_1, \ldots, i_N \leq d, \bar{h} \in \mathscr{B}_{\bar{M}}$,

$$
F^{i_1} \star \cdots \star F^{i_N}(\bar{h}) = J(F^{i_2} \star \cdots \star F^{i_N})(\bar{h}) F^{i_1}(\bar{h}). \tag{17}
$$

**Linear case.** We start with the proof of the linear case before moving on to the general case. When $\sigma$ is chosen to be the identity function, each $F_{\mathrm{RNN}}^i$ is an affine vector field, in the sense that $F_{\mathrm{RNN}}^i(\bar{h}) = W_i \bar{h} + b_i$, where $W_i = 0_{\bar{e} \times \bar{e}}$, $b_i$ is the $i + d$th vector of the canonical basis of $\mathbb{R}^{e+d}$, and

$$
W_{d+1} = \begin{pmatrix} \frac{2}{1-L} W \\ 0_{d \times \bar{e}} \end{pmatrix} \quad \text{and} \quad b_{d+1} = \begin{pmatrix} \frac{2}{1-L} b \\ 0_d \end{pmatrix}.
$$

Since $J(F_{\mathrm{RNN}}^i) = W_i$, we have, for any $\bar{h} \in \mathbb{R}^{e+d}$ and any $1 \leq i_1, \ldots, i_k \leq d$,

$$
F_{\mathrm{RNN}}^{i_1} \star \cdots \star F_{\mathrm{RNN}}^{i_k}(\bar{h}) = W_{i_k} \cdots W_{i_2}(W_{i_1} \bar{h} + b_{i_1}).
$$

Thus, for any $\bar{h} \in \mathscr{B}_{\bar{M}}$,

$$
\|F_{\mathrm{RNN}}^{i_1} \star \cdots \star F_{\mathrm{RNN}}^{i_k}(\bar{h})\| \leq \|W_{i_k}\|_{\mathrm{op}} \cdots \|W_{i_2}\|_{\mathrm{op}} (\|W_{i_1}\|_{\mathrm{op}} \bar{M} + \|b_{i_1}\|).
$$

For $i \neq d + 1$, $\|W_{i_1}\|_{\mathrm{op}} = 0$, and so

$$
\Lambda_k(\mathbf{F}_{\mathrm{RNN}}) \leq C \|W_{d+1}\|_{\mathrm{op}}^{k-1},
$$

with $C = \|W_{d+1}\|_{\mathrm{op}} \bar{M} + \max(1, 2(1-L)^{-1} \|b\|)$. Therefore, using the convention $\Lambda_0(\mathbf{F}_{\mathrm{RNN}}) = \bar{M}$,

$$
\sum_{k=0}^{\infty} \frac{d^k}{k!} \Lambda_k(\mathbf{F}_{\mathrm{RNN}}) \leq \bar{M} + Cd \sum_{k=1}^{\infty} \frac{1}{k!} \big( 2d(1-L)^{-1} \|W\|_{\mathrm{op}} \big)^{k-1} < \infty.
$$

**General case.** In the general case, the proof is two-fold. First, we upper bound (17) by a function of the norms of higher-order Jacobians of $F^{i_1}, \ldots, F^{i_N}$. We then apply this bound to the specific case $\mathbf{F} = \mathbf{F}_{\mathrm{RNN}}$. We refer to Appendix C for details on higher-order derivatives in tensor spaces. Let $F : \mathbb{R}^e \to \mathbb{R}^e$ be a smooth vector field. If $F(h) = (F_1(h), \ldots, F_e(h))^{\top}$, each of its coordinates $F_i$ is a function from $\mathbb{R}^e$ to $\mathbb{R}$, $\mathscr{C}^{\infty}$ with respect to all its input variables. We define the derivative of order $k$ of $F$ as the tensor field

$$
\begin{aligned}
J^k(F) : \mathbb{R}^e &\to (\mathbb{R}^e)^{\otimes k+1} \\
h &\mapsto J^k(F)(h),
\end{aligned}
$$

where

$$
J^k(F)(h) = \sum_{1 \leq j, i_1, \ldots, i_k \leq e} \frac{\partial^k F_j(h)}{\partial h_{i_1} \ldots \partial h_{i_k}} e_j \otimes e_{i_1} \otimes \cdots \otimes e_{i_k}.
$$

We take the convention $J^0(F) = F$, and note that $J(F) = J^1(F)$ is the Jacobian matrix, and that $J^k(J^{k'}(F)) = J^{k+k'}(F)$.

**Lemma 3.** *Let $A^1, \ldots, A^k : \mathbb{R}^e \to \mathbb{R}^e$ be smooth vector fields. Then, for any $h \in \mathbb{R}^e$*

$$\left\| A^k \star \cdots \star A^1(h) \right\| \leq \sum_{n_1 + \cdots + n_k = k-1} C(k; n_1, \ldots, n_k) \| J^{n_1}(A^1)(h) \| \cdots \| J^{n_k}(A^k)(h) \|,$$

*where $C(k; n_1, \ldots, n_k)$ is defined by the following recurrence on $k$: $C(1; 0) = 1$ and for any $n_1, \ldots, n_{k+1} \geq 0$,*

$$C(k+1; n_1, \ldots, n_{k+1}) = \sum_{\ell=1}^{k} C(k; n_1, \ldots, n_\ell - 1, \ldots, n_k) \qquad \text{if} \quad n_{k+1} = 0, \qquad (18)$$

$$C(k+1; n_1, \ldots, n_{k+1}) = 0 \qquad \qquad \text{otherwise.}$$

*Proof.* We refer to Appendix C for the definitions of the tensor dot product $\odot$ and tensor permutations, as well as for computation rules involving these operations. We show in fact by induction a stronger result, namely that there exist tensor permutations $\pi_p$ such that

$$A^k \star \cdots \star A^1(h) = \sum_{n_1 + \cdots + n_k = k-1} \sum_{1 \leq p \leq C(k; n_1, \ldots, n_k)} \pi_p \left[ J^{n_1}(A^1)(h) \odot \cdots \odot J^{n_k}(A^k)(h) \right]. \quad (19)$$

Note that we do not make explicit the permutations nor the axes of the tensor dot operations since we are only interested in bounding the norm of the iterated star products. Also, for simplicity, we denote all permutations by $\pi$, even though they may change from line to line.

We proceed by induction on $k$. For $k = 1$, the formula is clear. Assume that the formula is true at order $k$. Then

$$J(A^k \star \cdots \star A^1)$$
$$= \sum_{n_1 + \cdots + n_k = k-1} \sum_{1 \leq p \leq C(k; n_1, \ldots, n_k)} J\left[ \pi_p [ J^{n_1}(A^1) \odot \cdots \odot J^{n_k}(A^k) ] \right]$$
$$= \sum_{n_1 + \cdots + n_k = k-1} \sum_{1 \leq p \leq C(k; n_1, \ldots, n_k)} \pi_p \left[ J[ J^{n_1}(A^1) \odot \cdots \odot J^{n_k}(A^k) ] \right]$$
$$= \sum_{n_1 + \cdots + n_k = k-1} \sum_{1 \leq p \leq C(k; n_1, \ldots, n_k)} \sum_{\ell=1}^{k} \pi_p \circ \pi_\ell \Big[ J^{n_1}(A^1) \odot$$
$$\cdots \odot J^{n_\ell + 1}(A^\ell) \odot \cdots \odot J^{n_k}(A^k) \Big].$$

In the inner sum, we introduce the change of variable $p_i = n_i$ for $i \neq \ell$ and $p_\ell = n_\ell + 1$. This yields
$$J(A^k \star \cdots \star A^1)$$
$$= \sum_{p_1 + \cdots + p_k = k} \sum_{\ell=1}^{k} \sum_{1 \leq p \leq C(k; p_1, \ldots, p_\ell - 1, \ldots, p_k)} \pi_p \circ \pi_\ell \Big[ J^{n_1}(A^1) \odot$$
$$\cdots \odot J^{n_\ell + 1}(A^\ell) \odot \cdots \odot J^{n_k}(A^k) \Big]$$
$$= \sum_{p_1 + \cdots + p_{k+1} = k} \sum_{1 \leq q \leq C(k+1; p_1, \ldots, p_{k+1})} \pi_q \Big[ J^{n_1}(A^1) \odot \cdots \odot J^{p_k}(A^k) \Big],$$

where in the last sum the only non-zero term is for $p_{k+1} = 0$. To conclude the induction, it remains to note that

$$A^{k+1} \star \cdots \star A^1 = J(A^k \star \cdots \star A^1) \odot A^{k+1} = J(A^k \star \cdots \star A^1) \odot J^0(A^{k+1}).$$

Hence,
$$A^{k+1} \star \cdots \star A^1$$
$$= \sum_{p_1 + \cdots + p_{k+1} = k} \sum_{1 \leq q \leq C(k+1; p_1, \ldots, p_{k+1})} \pi_q \left[ J^{n_1}(A^1) \odot \cdots \odot J^{p_k}(A^k) \right] \odot J^{p_{k+1}}(A^{k+1})$$
$$= \sum_{p_1 + \cdots + p_{k+1} = k} \sum_{1 \leq q \leq C(k+1; p_1, \ldots, p_{k+1})} \pi_q \left[ J^{n_1}(A^1) \odot \cdots \odot J^{p_k}(A^k) \odot J^{p_{k+1}}(A^{k+1}) \right].$$

The result is then a consequence of (19) and of Lemma 6. $\qquad \square$

We now restrict ourselves to the case $\mathbf{F} = \mathbf{F}_{\mathrm{RNN}}$ as defined by (5) and give an upper bound on the higher-order derivatives of the tensor fields $F^{i_1}, \ldots, F^{i_N}$.

**Lemma 4.** *For any $i \in \{1, \ldots, d+1\}$, $\bar{h} \in \mathscr{B}_{\bar{M}}$, for any $k \geq 0$,*

$$\|J^k(F^i_{\mathrm{RNN}})(\bar{h})\| \leq \Big(\frac{2}{1-L}\|W\|_F\Big)^k \|\sigma^{(k)}\|_\infty.$$

*Proof.* For any $1 \leq i \leq d$, $F^i_{\mathrm{RNN}}(\bar{h})$ is constant, so $J^k(F^1_{\mathrm{RNN}}) = \cdots = J^k(F^d_{\mathrm{RNN}}) = 0$. For $i = d+1$, we have, for any $1 \leq j \leq e$,

$$\frac{\partial^k F^{d+1}_{\mathrm{RNN},j}(\bar{h})}{\partial \bar{h}_{i_1} \ldots \partial \bar{h}_{i_k}} = \Big(\frac{2}{1-L}\Big)^k W_{ji_1} \cdots W_{ji_k} \sigma^{(k)}(W_{j\cdot}\bar{h} + b),$$

where $W_{j\cdot}$ denotes the $j$th row of $W$ and for $e+1 \leq j \leq \bar{e}$, $F^{d+1}_j = 0$. Therefore,

$$
\begin{aligned}
\|J^k(F^{d+1}_{\mathrm{RNN}})(\bar{h})\|^2 &\leq \Big(\frac{2}{1-L}\Big)^{2k} \sum_{1 \leq j, i_1, \ldots, i_k \leq e} |W_{ji_1} \cdots W_{ji_k} \sigma^{(k)}(W_{j\cdot}\bar{h} + b)|^2 \\
&= \Big(\frac{2}{1-L}\Big)^{2k} \|\sigma^{(k)}\|^2_\infty \sum_j \Big(\sum_i |W_{ji}|^2\Big)^k \\
&\leq \Big(\frac{2}{1-L}\Big)^{2k} \|\sigma^{(k)}\|^2_\infty \|W\|^{2k}_F.
\end{aligned}
$$

$\square$

We are now in a position to conclude the proof using condition (11). By Lemma 3 and 4, for any $1 \leq i_1, \ldots, i_N \leq d+1$,

$$
\begin{aligned}
\big\|F^{i_1}_{\mathrm{RNN}} \star \cdots \star F^{i_N}_{\mathrm{RNN}}(\bar{h})\big\| \\
\leq \sum_{n_1 + \cdots + n_N = N-1} C(N; n_N, \ldots, n_1) \|J^{n_N}(F^{i_N}_{\mathrm{RNN}})(\bar{h})\| \cdots \|J^{n_1}(F^{i_1}_{\mathrm{RNN}})(\bar{h})\| \\
\leq \Big(\frac{2}{1-L}\|W\|_F\Big)^{N-1} \sum_{n_1 + \cdots + n_N = N-1} C(N; n_N, \ldots, n_1) a^{n_1+1} n_1! \cdots a^{n_N+1} n_N! \\
\leq a\Big(\frac{2}{1-L}a^2\|W\|_F\Big)^{N-1} \sum_{n_1 + \cdots + n_N = N-1} C(N; n_N, \ldots, n_1) n_1! \cdots n_N!.
\end{aligned}
$$

Assume for the moment that $C(N; n_N, \ldots, n_1)$ is smaller than the multinomial coefficient $\binom{N}{n_N, \ldots, n_1}$. Then, using the fact that there are $\binom{n+k-1}{k-1}$ weak compositions of $n$ in $k$ parts and Stirling's approximation, we have

$$
\begin{aligned}
\Lambda_N(\mathbf{F}) &\leq a\Big(\frac{2}{1-L}a^2\|W\|_F\Big)^{N-1} N! \times \mathrm{Card}(\{n_1 + \cdots + n_N = N-1\}) \\
&\leq a\Big(\frac{2}{1-L}a^2\|W\|_F\Big)^{N-1} N! \binom{2N-2}{N-1} \\
&\leq \frac{a}{2}\Big(\frac{2}{1-L}a^2\|W\|_F\Big)^{N-1} N! \binom{2N}{N} \\
&\leq a\frac{\sqrt{2}e}{\pi}\Big(\frac{8}{1-L}a^2\|W\|_F\Big)^{N-1} \frac{N!}{\sqrt{N}}.
\end{aligned}
$$

Hence, provided $\|W\|_F < (1-L)/8a^2 d$,

$$\sum_{k=1}^\infty \frac{d^k}{k!}\Lambda_k(\mathbf{F}) \leq ad\frac{\sqrt{2}e}{\pi} \sum_{k=1}^\infty \Big(\frac{8da^2\|W\|_F}{1-L}\Big)^{k-1} \frac{1}{\sqrt{k}} < \infty,$$

and $(A_2)$ is verified.

To conclude the proof, it remains to prove the following lemma.

**Lemma 5.** *For any $k \geq 1$ and $n_1, \ldots, n_k \geq 0$, $C(k; n_1, \ldots, n_k) \leq \binom{k-1}{n_1, \ldots, n_k}$.*

*Proof.* The proof is done by induction, by comparing the recurrence formula (18) with the following recurrence formula for multinomial coefficients:

$$\binom{k}{n_1, \ldots, n_{k+1}} = \sum_{\ell=1}^{k+1} \binom{k-1}{n_1, \ldots, n_\ell - 1, \ldots, n_{k+1}}.$$

More precisely, for $k = 1$, $C(1; 0) = 1 \leq \binom{0}{0} = 1$ and $C(1; 1) = 0 \leq \binom{0}{1} = 0$. Assume that the formula is true at order $k$. Then, at order $k+1$, there are two cases. If $n_{k+1} \neq 0$, $C(k+1; n_1, \ldots, n_{k+1}) = 0$, and the result is clear. On the other hand, if $n_{k+1} = 0$,

$$
\begin{aligned}
C(k+1; n_1, \ldots, n_k, 0) &= \sum_{\ell=1}^{k} C(k; n_1, \ldots, n_\ell - 1, \ldots, n_k) \\
&\leq \sum_{\ell=1}^{k} \binom{k-1}{n_1, \ldots, n_\ell - 1, \ldots, n_k} \\
&\leq \sum_{\ell=1}^{k+1} \binom{k-1}{n_1, \ldots, n_\ell - 1, \ldots, n_{k+1}} \\
&\leq \binom{k}{n_1, \ldots, n_{k+1}}.
\end{aligned}
$$

$\square$

## B.6 Proof of Theorem 1

First, Propositions 1 and 2 state that if $\bar{H}$ is the solution of (4) and Proj denotes the projection on the first $e$ coordinates, then

$$\left| z_T - \psi\big(\mathrm{Proj}(\bar{H}_1)\big) \right| = \left| \psi(h_T) - \psi\big(\mathrm{Proj}(\bar{H}_1)\big) \right| \leq \|\psi\|_{\mathrm{op}} \big\| h_T - \mathrm{Proj}(\bar{H}_1) \big\| \leq \|\psi\|_{\mathrm{op}} \frac{c_1}{T}.$$

For any $1 \leq k \leq N$, we let $\mathscr{D}^k(\bar{H}_0) : (\mathbb{R}^d)^{\otimes k} \to \mathbb{R}^e$ be the linear function defined by

$$\mathscr{D}^k(\bar{H}_0)(e_{i_1} \otimes \cdots \otimes e_{i_k}) = F^{i_1} \star \cdots \star F^{i_k}(\bar{H}_0), \tag{20}$$

where $e_1, \ldots, e_d$ denotes the canonical basis of $\mathbb{R}^{\bar{d}}$. We take the convention $(\mathbb{R}^d)^{\otimes 0} = \mathbb{R}$ and $\mathscr{D}^0(\bar{H}_0)(x) = \bar{H}_0$ for any $x \in \mathbb{R}$. Then, under assumptions $(A_1)$ and $(A_2)$, if $\bar{\mathbb{X}}^k$ denotes the signature of order $k$ of the path $\bar{X}_t = (X_t^\top, \frac{1-L}{2} t)^\top$, according to Propositions 4 and 5,

$$\bar{H}_1 = \bar{H}_0 + \sum_{k=1}^{\infty} \frac{1}{k!} \sum_{1 \leq i_1, \ldots, i_k \leq d} S_{[0,t]}^{(i_1, \ldots, i_k)}(X) F^{i_1} \star \cdots \star F^{i_k}(\bar{H}_0) = \sum_{k=0}^{\infty} \frac{1}{k!} \mathscr{D}^k(\bar{H}_0)(\mathbb{X}_{[0,t]}^k),$$

and

$$\psi \circ \mathrm{Proj}(\bar{H}_1) = \psi \circ \mathrm{Proj}\Big( \sum_{k=0}^{\infty} \frac{1}{k!} \mathscr{D}^k(\bar{H}_0)(\bar{\mathbb{X}}^k) \Big) = \sum_{k=0}^{\infty} \frac{1}{k!} \psi \circ \mathrm{Proj}\big( \mathscr{D}^k(\bar{H}_0)(\bar{\mathbb{X}}^k) \big),$$

by linearity of $\psi$ and Proj. Since the maps $\mathscr{D}^k(\bar{H}_0) : (\mathbb{R}^d)^{\otimes k} \to \mathbb{R}^e$ are linear, the above equality takes the form

$$\psi \circ \mathrm{Proj}(\bar{H}_1) = \sum_{k=0}^{\infty} \langle \alpha^k, \bar{\mathbb{X}}^k \rangle_{(\mathbb{R}^d)^{\otimes k}}, \tag{21}$$

where $\alpha^k \in (\mathbb{R}^d)^{\otimes k}$ is the coefficient of the linear map $\frac{1}{k!}\psi \circ \mathrm{Proj} \circ \mathscr{D}^k(\bar{H}_0)$ in the canonical basis. Let $\alpha = (\alpha^0, \ldots, \alpha^k, \ldots)$. Under assumption $(A_2)$,

$$\sum_{k=0}^{\infty} \|\alpha^k\|^2_{(\mathbb{R}^d)^{\otimes k}} \leq \sum_{k=0}^{\infty} \sum_{1 \leq i_1, \ldots, i_k \leq d} \left(\frac{1}{k!}\right)^2 \|\psi\|^2_{\mathrm{op}} \|F^{i_1} \star \cdots \star F^{i_k}(\bar{H}_0)\|^2$$

$$\leq \|\psi\|^2_{\mathrm{op}} \sum_{k=0}^{\infty} \sum_{1 \leq i_1, \ldots, i_k \leq d} \left(\frac{1}{k!}\right)^2 \Lambda_k(\mathbf{F})^2$$

$$\leq \|\psi\|^2_{\mathrm{op}} \sum_{k=0}^{\infty} \left(\frac{d^k}{k!} \Lambda_k(\mathbf{F})\right)^2 < \infty.$$

This shows that $\alpha \in \mathscr{T}$, and therefore, using (21), we conclude

$$\|z_T - \langle \alpha, S(\bar{X})\rangle_{\mathscr{T}}\| \leq \|\psi\|_{\mathrm{op}} \frac{c_1}{T}.$$

## B.7   Proof of Theorem 2

Let

$$\mathscr{G} = \left\{ g_\theta : (\mathbb{R}^d)^T \to \mathbb{R} \mid g_\theta(\mathbf{x}) = z_T, \theta \in \Theta \right\}$$

be the function class of (discrete) RNN and

$$\mathscr{S} = \left\{ \xi_{\alpha_\theta} : \mathscr{X} \to \mathbb{R} \mid \xi_{\alpha_\theta}(X) = \langle \alpha_\theta, S(\bar{X})\rangle_{\mathscr{T}}, \theta \in \Theta \right\},$$

be the class of their RKHS embeddings, where $\alpha_\theta$ is defined by (21). For any $\theta \in \Theta$, we let

$$\mathscr{R}_{\mathscr{G}}(\theta) = \mathbb{E}[\ell(\mathbf{y}, g_\theta(\mathbf{x}))], \quad \text{and} \quad \mathscr{R}_{\mathscr{S}}(\theta) = \mathbb{E}[\ell(\mathbf{y}, \xi_{\alpha_\theta}(\bar{X}))],$$

and denote by $\widehat{\mathscr{R}}_{n,\mathscr{G}}$ and $\widehat{\mathscr{R}}_{n,\mathscr{S}}$ the corresponding empirical risks. We also let $\theta^*_{\mathscr{G}}$, $\theta^*_{\mathscr{S}}$, $\widehat{\theta}_{n,\mathscr{G}}$, and $\widehat{\theta}_{n,\mathscr{S}}$ be the corresponding minimizers. We have

$$\mathbb{P}\left(\mathbf{y} g_{\widehat{\theta}_n}(\mathbf{x}) \leq 0 \mid \mathscr{D}_n\right) - \widehat{\mathscr{R}}_{n,\mathscr{G}}(\widehat{\theta}_{n,\mathscr{G}}) \leq \mathbb{E}\left[\ell(\mathbf{y}, g_{\widehat{\theta}_{n,\mathscr{G}}}(\mathbf{x}))\right] - \widehat{\mathscr{R}}_{n,\mathscr{G}}(\widehat{\theta}_{n,\mathscr{G}})$$

$$= \mathscr{R}_{\mathscr{G}}(\widehat{\theta}_{n,\mathscr{G}}) - \widehat{\mathscr{R}}_{n,\mathscr{G}}(\widehat{\theta}_{n,\mathscr{G}})$$

$$= \mathscr{R}_{\mathscr{G}}(\widehat{\theta}_{n,\mathscr{G}}) - \mathscr{R}_{\mathscr{S}}(\widehat{\theta}_{n,\mathscr{G}}) + \mathscr{R}_{\mathscr{S}}(\widehat{\theta}_{n,\mathscr{G}}) - \widehat{\mathscr{R}}_{n,\mathscr{S}}(\widehat{\theta}_{n,\mathscr{G}})$$

$$+ \widehat{\mathscr{R}}_{n,\mathscr{S}}(\widehat{\theta}_{n,\mathscr{G}}) - \widehat{\mathscr{R}}_{n,\mathscr{G}}(\widehat{\theta}_{n,\mathscr{G}})$$

$$\leq \sup_\theta |\mathscr{R}_{\mathscr{G}}(\theta) - \mathscr{R}_{\mathscr{S}}(\theta)| + \sup_\theta |\mathscr{R}_{\mathscr{S}}(\theta) - \widehat{\mathscr{R}}_{n,\mathscr{S}}(\theta)|$$

$$+ \sup_\theta |\widehat{\mathscr{R}}_{n,\mathscr{G}}(\theta) - \widehat{\mathscr{R}}_{n,\mathscr{S}}(\theta)|.$$

Using Theorem 1, we have

$$\sup_\theta |\mathscr{R}_{\mathscr{G}}(\theta) - \mathscr{R}_{\mathscr{S}}(\theta)| = \sup_\theta |\mathbb{E}\left[\ell(\mathbf{y}, g_\theta(\mathbf{x})) - \ell(\mathbf{y}, \xi_{\alpha_\theta}(\bar{X}))\right]|$$

$$\leq \sup_\theta \mathbb{E}\left[|\phi(\mathbf{y} g_\theta(\mathbf{x})) - \phi(\mathbf{y} \xi_{\alpha_\theta}(\bar{X}))|\right]$$

$$\leq \sup_\theta \mathbb{E}\left[K_\ell |\mathbf{y}| \times |g_\theta(\mathbf{x}) - \xi_{\alpha_\theta}(\bar{X})|\right]$$

$$\leq K_\ell \sup_\theta (\|\psi\|_{\mathrm{op}} c_{1,\theta}) \frac{1}{T} := \frac{c_2}{2T},$$

where $c_{1,\theta} = K_{f_\theta} e^{K_{f_\theta}} \left(L + \|f_\theta\|_\infty e^{K_{f_\theta}}\right)$ (the infinity norm $\|f_\theta\|_\infty$ is taken on the balls $\mathscr{B}_L$ and $\mathscr{B}_M$). One proves with similar arguments that

$$\sup_\theta |\widehat{\mathscr{R}}_{n,\mathscr{G}}(\theta) - \widehat{\mathscr{R}}_{n,\mathscr{S}}(\theta)| \leq \frac{c_2}{2T}.$$

Under the assumption of the theorem, there exists a ball $\mathscr{B} \subset \mathscr{H}$ of radius $B$ such that $\mathscr{S} \subset \mathscr{B}$. This yields

$$\sup_{\theta}|\mathscr{R}_{\mathscr{S}}(\theta) - \widehat{\mathscr{R}}_{n,\mathscr{S}}(\theta)| \leq \sup_{\alpha \in \mathscr{T}, \|\alpha\|_{\mathscr{T}} \leq B}|\mathscr{R}_{\mathscr{B}}(\alpha) - \widehat{\mathscr{R}}_{n,\mathscr{B}}(\alpha)|,$$

where

$$\mathscr{R}_{\mathscr{B}}(\alpha) = \mathbb{E}[\ell(Y, \xi_{\alpha}(\bar{X}))] \quad \text{and} \quad \widehat{\mathscr{R}}_{n,\mathscr{B}}(\alpha) = \frac{1}{n}\sum_{i=1}^{n}\ell(Y^{(i)}, \xi_{\alpha}(\bar{X}^{(i)})).$$

We now have reached a familiar situation where the supremum is over a ball in an RKHS. A slight extension of Bartlett and Mendelson (2002, Theorem 8) yields that with probability at least $1 - \delta$,

$$\sup_{\alpha \in \mathscr{T}, \|\alpha\|_{\mathscr{T}} \leq B}|\mathscr{R}_{\mathscr{B}}(\alpha) - \widehat{\mathscr{R}}_{n,\mathscr{B}}(\alpha)| \leq 4K_{\ell}\mathbb{E}\mathrm{Rad}_n(\mathscr{B}) + 2BK_{\ell}(1-L)^{-1}\sqrt{\frac{\log(1/\delta)}{2n}},$$

where $\mathrm{Rad}_n(\mathscr{B})$ denotes the Rademacher complexity of $\mathscr{B}$. Observe that we have used the fact that the loss is bounded by $2BK_{\ell}(1-L)^{-1}$ since, for any $\xi_{\alpha} \in \mathscr{B}$, by the Cauchy-Schwartz inequality,

$$\ell(\mathbf{y}, \xi_{\alpha}(\bar{X})) = \phi(\mathbf{y}\langle \alpha, S(\bar{X})\rangle_{\mathscr{T}}) \leq K_{\ell}|\mathbf{y}\langle \alpha, S(\bar{X})\rangle_{\mathscr{T}}| \leq K_{\ell}\|\alpha\|_{\mathscr{T}}\|S(\bar{X})\|_{\mathscr{T}}$$
$$\leq 2K_{\ell}B(1-L)^{-1}.$$

Finally, the proof follows by noting that Rademacher complexity of $\mathscr{B}$ is bounded by

$$\mathrm{Rad}_n(\mathscr{B}) \leq \frac{B}{n}\sqrt{\sum_{i=1}^{n}K(X^{(i)}, X^{(i)})} = \frac{B}{n}\sqrt{\sum_{i=1}^{n}\|S(\bar{X}^{(i)})\|_{\mathscr{T}}^2} \leq \frac{2B(1-L)^{-1}}{\sqrt{n}}.$$

## B.8 Proof of Theorem 3

Let

$$\mathscr{G} = \left\{g_{\theta} : (\mathbb{R}^d)^T \to (\mathbb{R}^p)^T \mid g_{\theta}(\mathbf{x}) = (z_1, \ldots, z_T), \theta \in \Theta\right\}$$

be the function class of discrete RNN in a sequential setting. Let

$$\mathscr{S} = \left\{\Gamma_{\theta} : \mathscr{X} \to (\mathbb{R}^p)^T \mid \Gamma_{\theta}(X) = \left(\Xi_{\theta}(\tilde{X}_{[1]}), \ldots, \Xi_{\theta}(\tilde{X}_{[T]})\right)\right\},$$

be the class of their RKHS embeddings, where $\tilde{X}_{[j]}$ is the path equal to $X$ on $[0, j/T]$ and then constant on $[j/T, 1]$ (see Figure 4). For any $X \in \mathscr{X}$,

$$\Xi_{\theta}(a) = \begin{pmatrix} \langle \alpha_{1,\theta}, S(\bar{X})\rangle_{\mathscr{T}} \\ \vdots \\ \langle \alpha_{p,\theta}, S(\bar{X})\rangle_{\mathscr{T}} \end{pmatrix} = \begin{pmatrix} \xi_{\alpha_{1,\theta}}(X) \\ \vdots \\ \xi_{\alpha_{p,\theta}}(X) \end{pmatrix} \in \mathbb{R}^p,$$

where $(\alpha_{1,\theta}, \ldots, \alpha_{p,\theta})^{\top} \in (\mathscr{T})^p$ are the coefficients of the linear maps $\frac{1}{k!}\psi \circ \mathrm{Proj} \circ \mathscr{D}^k(\bar{H}_0) : (\mathbb{R}^d)^{\otimes k} \to \mathbb{R}^p$, $k \geq 0$, in the canonical basis, where $\mathscr{D}^k$ is defined by (20).

We start the proof as in Theorem 2, until we obtain

$$\mathscr{R}_{\mathscr{G}}(\widehat{\theta}_{n,\mathscr{G}}) - \widehat{\mathscr{R}}_{n,\mathscr{G}}(\widehat{\theta}_{n,\mathscr{G}}) \leq \sup_{\theta}|\mathscr{R}_{\mathscr{G}}(\theta) - \mathscr{R}_{\mathscr{S}}(\theta)| + \sup_{\theta}|\mathscr{R}_{\mathscr{S}}(\theta) - \widehat{\mathscr{R}}_{n,\mathscr{S}}(\theta)|$$
$$+ \sup_{\theta}|\widehat{\mathscr{R}}_{n,\mathscr{G}}(\theta) - \widehat{\mathscr{R}}_{n,\mathscr{S}}(\theta)|.$$

By definition of the loss, for any $\theta \in \Theta$,

$$|\mathscr{R}_{\mathscr{G}}(\theta) - \mathscr{R}_{\mathscr{S}}(\theta)| = \left|\mathbb{E}\big[\ell(\mathbf{y}, g_\theta(\mathbf{x})) - \ell(\mathbf{y}, \Gamma_\theta(X))\big]\right|$$

$$\leq \mathbb{E}\Big[\big|\frac{1}{T}\sum_{j=1}^{T}\big(\|y_j - z_j\|^2 - \|y_j - \Xi_\theta(\tilde{X}_{[j]})\|^2\big)\big|\Big]$$

$$\leq \mathbb{E}\Big[\frac{1}{T}\sum_{j=1}^{T}\big|\langle z_j + \Xi_\theta(\tilde{X}_{[j]}) - 2y_j, z_j - \Xi_\theta(\tilde{X}_{[j]})\rangle\big|\Big]$$

$$\leq \mathbb{E}\Big[\frac{1}{T}\sum_{j=1}^{T}\|z_j + \Xi_\theta(\tilde{X}_{[j]}) - 2y_j\| \times \|z_j - \Xi_\theta(\tilde{X}_{[j]})\|\Big]$$

(by the Cauchy-Schwartz inequality).

According to inequality (14), one has

$$\|z_j - \Xi_\theta(\tilde{X}_{[j]})\| \leq \|\psi\|_{\mathrm{op}}\frac{c_{1,\theta}}{T},$$

where $c_{1,\theta} = K_{f_\theta}e^{K_{f_\theta}}\big(L + \|f_\theta\|_\infty e^{K_{f_\theta}}\big)$. Moreover,

$$\left\|\Xi_\theta(\tilde{X}_{[j]})\right\|^2 = \sum_{\ell=1}^{p}\big|\langle\alpha_{\ell,\theta}, S(\tilde{X}_{[j]})\rangle_{\mathscr{T}}\big|^2 \leq \sum_{\ell=1}^{p}\|\alpha_{\ell,\theta}\|_{\mathscr{T}}^2\|S(\tilde{X}_{[j]})\|_{\mathscr{T}}^2 \leq pB^2\big(2(1-L)^{-1}\big)^2,$$

since $\|S(\tilde{X}_{[j]})\|_{\mathscr{T}} = \|S_{[0,j/T]}(\bar{X})\|_{\mathscr{T}} \leq \|S(\bar{X})\|_{\mathscr{T}}$. This yields

$$\|z_j + \Xi_\theta(\tilde{X}_{[j]}) - 2y_j\| \leq \|z_j\| + \|\Xi_\theta(\tilde{X}_{[j]})\| + 2\|y_j\|$$
$$\leq \|\psi\|_{\mathrm{op}}\|f_\theta\|_\infty + 2\sqrt{p}B(1-L)^{-1} + 2K_y.$$

Finally,

$$\sup_\theta|\mathscr{R}_{\mathscr{G}}(\theta) - \mathscr{R}_{\mathscr{S}}(\theta)| \leq \frac{c_3}{2T},$$

where $c_3 = \sup_\theta\big(c_{1,\theta} + \|\psi\|_{\mathrm{op}}\|f_\theta\|_\infty\big) + 2\sqrt{p}B(1-L)^{-1} + 2K_y$. One proves with similar arguments that

$$\sup_\theta|\widehat{\mathscr{R}}_{n,\mathscr{G}}(\theta) - \widehat{\mathscr{R}}_{n,\mathscr{S}}(\theta)| \leq \frac{c_3}{2T}.$$

We now turn to the term $\sup_\theta|\mathscr{R}_{\mathscr{S}}(\theta) - \widehat{\mathscr{R}}_{n,\mathscr{S}}(\theta)|$. We have

$$\mathscr{R}_{\mathscr{S}}(\theta) - \widehat{\mathscr{R}}_{n,\mathscr{S}}(\theta)$$

$$= \mathbb{E}[\ell(\mathbf{y}, \Gamma_\theta(X))] - \frac{1}{n}\sum_{i=1}^{n}\ell(\mathbf{y}^{(i)}, \Gamma_\theta(X^{(i)}))$$

$$= \frac{1}{T}\sum_{j=1}^{T}\Big(\mathbb{E}[\|y_j - \Xi_\theta(\tilde{X}_{[j]})]\|^2 - \frac{1}{n}\sum_{i=1}^{n}\big\|y_j^{(i)} - \Xi_\theta(\tilde{X}_{[j]}^{(i)})\big\|^2\Big).$$

Therefore,

$$\sup_\theta|\mathscr{R}_{\mathscr{S}}(\theta) - \widehat{\mathscr{R}}_{n,\mathscr{S}}(\theta)| \leq \frac{1}{T}\sum_{j=1}^{T}\sup_\theta\Big|\mathbb{E}[\|y_j - \Xi_\theta(\tilde{X}_{[j]})]\|^2 - \frac{1}{n}\sum_{i=1}^{n}\big\|y_j^{(i)} - \Xi_\theta(\tilde{X}_{[j]}^{(i)})\big\|^2\Big|.$$

Note that for a fixed $j$, the pairs $(\tilde{X}_{[j]}^{(i)}, y_j^{(i)})$ are i.i.d. Under the assumptions of the theorem, there exists a ball $\mathscr{B} \subset \mathscr{H}$ such that for any $1 \leq \ell \leq p$, $\theta \in \Theta$, $\xi_{\alpha_{\ell,\theta}} \in \mathscr{B}$. We denote by $\mathscr{B}_p$ the sum of $p$ such spaces, that is,

$$\mathscr{B}_p = \big\{f_\alpha : \mathscr{X} \to \mathbb{R}^p \mid f_\alpha(X) = (f_{\alpha_1}(X), \ldots, f_{\alpha_p}(X))^\top, f_{\alpha_\ell} \in \mathscr{B}\big\}.$$

Clearly, $\Xi_\theta \in \mathscr{B}_p$, and it follows that

$$\sup_\theta \left| \mathbb{E}[\|y_j - \Xi_\theta(\tilde{X}_{[j]})]\|^2 - \frac{1}{n}\sum_{i=1}^n \left\| y_j^{(i)} - \Xi_\theta(\tilde{X}_{[j]}^{(i)}) \right\|^2 \right|$$

$$\leq \sup_{f_\alpha \in \mathscr{B}_p} \left| \mathbb{E}\big[\|y_j - f_\alpha(\tilde{X}_{[j]})\|^2\big] - \frac{1}{n}\sum_{i=1}^n \|y_j^{(i)} - f_\alpha(\tilde{X}_{[j]}^{(i)})\|^2 \right|.$$

We have once again reached a familiar situation, which can be dealt with by an easy extension of Bartlett and Mendelson (2002, Theorem 12). For any $f_\alpha \in \mathscr{B}_p$, let $\tilde{\phi} \circ f_\alpha : \mathscr{X} \times \mathbb{R}^p : (X, y) \mapsto \|y - f_\alpha(X)\|^2 - \|y\|^2$. Then, $\tilde{\phi} \circ f_\alpha$ is upper bounded by

$$|\tilde{\phi} \circ f_\alpha(X, y)| = \big| \|y - f_\alpha(X)\|^2 - \|y\|^2 \big| \leq \|f_\alpha(X)\|\big(\|f_\alpha(X)\| + 2\|y\|\big)$$
$$\leq 2\sqrt{p}B(1-L)^{-1}(2\sqrt{p}B(1-L)^{-1} + 2K_y)$$
$$\leq 4pB(1-L)^{-1}(B(1-L)^{-1} + K_y).$$

Let $c_4 = B(1-L)^{-1} + K_y$ and $c_5 = 4pB(1-L)^{-1}c_4 + K_y^2$. Then with probability at least $1 - \delta$,

$$\sup_{f_\alpha \in \mathscr{B}_p} \left| \mathbb{E}\big[\|y_j - f_\alpha(\tilde{X}_{[j]})\|\big] - \frac{1}{n}\sum_{i=1}^n \|y_j^{(i)} - f_\alpha(\tilde{X}_{[j]}^{(i)})\| \right| \leq \mathrm{Rad}_n(\tilde{\phi} \circ \mathscr{B}_p) + \sqrt{\frac{2c_5 \log(1/\delta)}{n}},$$

where $\tilde{\phi} \circ \mathscr{B}_p = \big\{ (X, y) \mapsto \tilde{\phi} \circ f_\alpha(X, y) | f_\alpha \in \mathscr{B}_p \big\}$. Elementary computations on Rademacher complexities yield

$$\mathrm{Rad}_n(\tilde{\phi} \circ \mathscr{B}_p) \leq 2pc_4 \mathrm{Rad}_n(\mathscr{B}) \leq \frac{4pc_4 B(1-L)^{-1}}{\sqrt{n}},$$

which concludes the proof.

# C   Differentiation with higher-order tensors

## C.1   Definition

We define the generalization of matrix product between square tensors of order $k$ and $\ell$.

**Definition 4.** *Let $a \in (\mathbb{R}^e)^{\otimes k}$, $b \in (\mathbb{R}^e)^{\otimes \ell}$, $p \in \{1, \ldots, k\}$, $q \in \{1, \ldots, \ell\}$. Then the tensor dot product along $(p, q)$, denoted by $a \odot_{p,q} b \in (\mathbb{R}^e)^{\otimes(k+\ell-2)}$, is defined by*

$$(a \odot_{p,q} b)_{(i_1,\ldots,i_{k-1},j_1,\ldots,j_{\ell-1})} = \sum_{j=1}^e a_{(i_1,\ldots,i_{p-1},j,i_p,\ldots,i_{k-1})} b_{(j_1,\ldots,j_{q-1},j,j_q,\ldots,j_{\ell-1})}.$$

This operation just consists in computing $a \otimes b$, and then summing the $p$th coordinate of $a$ with the $q$th coordinate of $b$. The $\odot$ operator is not associative. To simplify notation, we take the convention that it is evaluated from left to right, that is, we write $a \odot b \odot c$ for $(a \odot b) \odot c$.

**Definition 5.** *Let $a \in (\mathbb{R}^e)^{\otimes k}$. For a given permutation $\pi$ of $\{1, \ldots, k\}$, we denote by $\pi(a)$ the permuted tensor in $(\mathbb{R}^e)^{\otimes k}$ such that*

$$\pi(a)_{(i_1,\ldots,i_k)} = a_{(i_{\Pi(1)},\ldots,i_{\Pi(k)})}.$$

**Example 5.** *If $A$ is a matrix, then $A^T = \pi(A)$, with $\pi$ defined by $\pi(1) = 2, \pi(2) = 1$.*

## C.2   Computation rules

We need to obtain two computation rules for the tensor dot product: bounding the norm (Lemma 6) and differentiating (Lemma 7).

**Lemma 6.** *Let $a \in (\mathbb{R}^e)^{\otimes k}$, $b \in (\mathbb{R}^e)^{\otimes \ell}$. Then, for all $p, q$,*

$$\|a \odot_{p,q} b\|_{(\mathbb{R}^e)^{\otimes k+\ell-2d}} \leq \|a\|_{(\mathbb{R}^e)^{\otimes k}} \|b\|_{(\mathbb{R}^e)^{\otimes \ell}}.$$

*Proof.* By the Cauchy-Schwartz inequality,

$$\|a \odot_{p,q} b\|^2_{(\mathbb{R}^e)^{\otimes k+\ell-2}}$$

$$= \sum_{1 \leq i_1,\ldots,i_{k-1},j_1,\ldots,j_{\ell-1} \leq e} (a \odot_{p,q} b)^2_{(i_1,\ldots,i_{k-1},j_1,\ldots,j_{\ell-1})}$$

$$= \sum_{1 \leq i_1,\ldots,i_{k-1},j_1,\ldots,j_{\ell-1} \leq e} \Big( \sum_{1 \leq j \leq e} a_{(i_1,\ldots,i_{p-1},j,i_p,\ldots,i_{k-1})} b_{(j_1,\ldots,j_{q-1},j,j_q,\ldots,j_{\ell-1})} \Big)^2$$

$$\leq \sum_{i_1,\ldots,i_{k-1},j_1,\ldots,j_{\ell-1}} \Big( \sum_j a^2_{(i_1,\ldots,i_{p-1},j,i_p,\ldots,i_{k-1})} \Big) \Big( \sum_j b^2_{(j_1,\ldots,j_{q-1},j,j_q,\ldots,j_{\ell-1})} \Big)$$

$$\leq \sum_{i_1,\ldots,i_{k-1},j} a^2_{(i_1,\ldots,i_{p-1},j,i_p,\ldots,i_{k-1})} \sum_{j_1,\ldots,j_{\ell-1},j} b^2_{(j_1,\ldots,j_{q-1},j,j_q,\ldots,j_{\ell-1})}$$

$$\leq \|a\|^2_{(\mathbb{R}^e)^{\otimes k}} \|b\|^2_{(\mathbb{R}^e)^{\otimes \ell}}.$$

$\square$

**Lemma 7.** *Let $A : \mathbb{R}^e \to (\mathbb{R}^e)^{\otimes k}$, $B : \mathbb{R}^e \to (\mathbb{R}^e)^{\otimes \ell}$ be smooth vector fields, $p \in \{1,\ldots,k\}$, $q \in \{1,\ldots,\ell\}$. Let $A \odot_{p,q} B : \mathbb{R}^e \to (\mathbb{R}^e)^{\otimes k+\ell-2}$ be defined by $A \odot_{p,q} B(h) = A(h) \odot_{p,q} B(h)$. Then there exists a permutation $\pi$ such that*

$$J(A \odot_{p,q} B) = \pi(J(A) \odot_{p,q} B) + A \odot_{p,q} J(B).$$

*Proof.* The left-hand side takes the form

$$(J(A \odot_{p,q} B))_{i_1,\ldots,i_{k-1},j_1,\ldots,j_{\ell-1},m} = \sum_j \Big[ \frac{\partial A}{\partial h_m}_{(i_1,\ldots,i_{p-1},j,i_p,\ldots,i_{k-1})} B_{(j_1,\ldots,j_{q-1},j,j_q,\ldots,j_{\ell-1})}$$

$$+ A_{(i_1,\ldots,i_{p-1},j,i_p,\ldots,i_{k-1})} \frac{\partial B}{\partial h_m}_{(j_1,\ldots,j_{q-1},j,j_q,\ldots,j_{\ell-1})} \Big].$$

The first term of the right-hand side writes

$$(J(A) \odot_{p,q} B)_{i_1,\ldots,i_{k-1},m,j_1,\ldots,j_{\ell-1}} = \sum_j \Big[ \frac{\partial A}{\partial h_m}_{(i_1,\ldots,i_{p-1},j,i_p,\ldots,i_{k-1})} B_{(j_1,\ldots,j_{q-1},j,j_q,\ldots,j_{\ell-1})} \Big],$$

and the second one

$$(A \odot_{p,q} J(B))_{i_1,\ldots,i_{k-1},j_1,\ldots,j_{\ell-1},m} = \sum_j \Big[ A_{(i_1,\ldots,i_{p-1},j,i_p,\ldots,i_{k-1})} \frac{\partial B}{\partial h_m}_{(j_1,\ldots,j_{q-1},j,j_q,\ldots,j_{\ell-1})} \Big].$$

Let us introduce the permutation $\pi$ which keeps the first $(k-1)$ axes unmoved, and rotates the remaining $\ell$ ones such that the last axis ends up in $k$th position. Then

$$\pi(J(A) \odot_{p,q} B)_{i_1,\ldots,i_{k-1},j_1,\ldots,j_{\ell-1},m} = \sum_j \Big[ \frac{\partial A}{\partial h_m}_{(i_1,\ldots,i_{p-1},j,i_p,\ldots,i_{k-1})} B_{(j_1,\ldots,j_{q-1},j,j_q,\ldots,j_{\ell-1})} \Big].$$

Hence $J(A \odot_{p,q} B) = \pi(J(A) \odot_{p,q} B) + A \odot_{p,q} J(B)$, which concludes the proof. $\square$

The following two lemmas show how to compose the Jacobian and the tensor dot operations with permutations. Their proofs follow elementary operations and are therefore omitted.

**Lemma 8.** *Let $A : \mathbb{R}^e \to (\mathbb{R}^e)^{\otimes k}$ and $\pi$ a permutation of $\{1,\ldots,k\}$. Then there exists a permutation $\tilde{\pi}$ of $\{1,\ldots,k+1\}$ such that*

$$J(\pi(A)) = \tilde{\pi}(J(A)).$$

**Lemma 9.** *Let $a \in (\mathbb{R}^e)^{\otimes k}$, $b \in (\mathbb{R}^e)^{\otimes \ell}$, $p \in \{1,\ldots,k\}$, $q \in \{1,\ldots,\ell\}$, $\pi$ a permutation of $\{1,\ldots,k\}$. Then there exists $\tilde{p} \in \{1,\ldots,k\}$, $\tilde{q} \in \{1,\ldots,\ell\}$, and a permutation $\tilde{\pi}$ of $\{1,\ldots,k+\ell-2\}$ such that*

$$\pi(a) \odot_{p,q} b = \tilde{\pi}(a \odot_{\tilde{p},\tilde{q}} b).$$

The following result is a generalization of Lemma 7 to the case of a dot product of several tensors.

**Lemma 10.** *For $\ell \in \{1, \ldots, k\}$, $n_\ell \in \mathbb{N}$, let $A_\ell : \mathbb{R}^e \to (\mathbb{R}^e)^{\otimes n_\ell}$ be smooth tensor fields. For any $(p_\ell)_{1 \leq \ell \leq k-1}$ and $(q_\ell)_{1 \leq \ell \leq k-1}$ such that $p_\ell \in \{1, \ldots, n_\ell\}$, $q_\ell \in \{1, \ldots, n_{\ell+1}\}$, there exist $k$ permutations $(\pi_\ell)_{1 \leq \ell \leq k}$ such that*

$$J(A_1 \odot_{p_1,q_1} A_2 \odot_{p_2,q_2} \cdots \odot_{p_{k-1},q_{k-1}} A_k) = \sum_{\ell=1}^{k} \pi_\ell \left[ A_1 \odot A_2 \odot \cdots \odot J(A_\ell) \odot \cdots \odot A_k \right],$$

*where the dot products of the right-hand side are along some axes that are not specify for simplicity.*

*Proof.* The proof is done by induction on $k$. The formula for $k = 1$ is straightforward. Assume that the formula is true at order $k$. As before, we do not specify indexes for tensor dot products as we are only interested in their existence. By Lemma 9, we have

$$
\begin{aligned}
&J(A_1 \odot \cdots \odot A_{k+1}) \\
&= J((A_1 \odot \cdots \odot A_k) \odot A_{k+1}) \\
&= \pi(J(A_1 \odot \cdots \odot A_k) \odot A_{k+1}) + A_1 \odot \cdots \odot A_k \odot J(A_{k+1}) \\
&= \pi \left[ \sum_{\ell=1}^{k} \pi_\ell \left[ A_1 \odot A_2 \odot \cdots \odot J(A_\ell) \odot \cdots \odot A_k \right] \odot A_{k+1} \right] + A_1 \odot \cdots \odot A_k \odot J(A_{k+1}) \\
&= \pi \left[ \sum_{\ell=1}^{k} \tilde{\pi}_\ell \left[ A_1 \odot A_2 \odot \cdots \odot J(A_\ell) \odot \cdots \odot A_k \odot A_{k+1} \right] \right] + A_1 \odot \cdots \odot A_k \odot J(A_{k+1}) \\
&= \sum_{\ell=1}^{k} \hat{\pi}_\ell \left[ A_1 \odot A_2 \odot \cdots \odot J(A_\ell) \odot \cdots \odot A_k \odot A_{k+1} \right] + A_1 \odot \cdots \odot A_k \odot J(A_{k+1}) \\
&\quad (\text{where } \hat{\pi} = \pi \circ \tilde{\pi}) \\
&= \sum_{\ell=1}^{k+1} \hat{\pi}_\ell \left[ A_1 \odot A_2 \odot \cdots \odot J(A_\ell) \odot \cdots \odot A_k \odot A_{k+1} \right].
\end{aligned}
$$

$\square$

# D  Experimental details

All the code to reproduce the experiments is available on GitHub at https://github.com/afermanian/rnn-kernel. Our experiments are based on the PyTorch (Paszke et al., 2019) framework. When not specified, the default parameters of PyTorch are used.

**Convergence of the Taylor expansion.**   For Figure 1, $10^3$ random RNN with 2 hidden units are generated, with the default weight initialization. The activation is either the logistic or the hyperbolic tangent. In Figure 1b, only the results with the logistic activation are plotted. The process $X$ is taken as a 2-dimensional spiral. The reference solution to the ODE (3) is computed with a numerical integration method from SciPy (Virtanen et al., 2020, scipy.integrate.solve_ivp with the 'LSODA' method). The signature in the step-$N$ Taylor expansion is computed with the package Signatory (Kidger and Lyons, 2021).

The step-$N$ Taylor expansion requires computing higher-order derivatives of tensor fields (up to order $N$). This is a highly non-trivial task since standard deep learning frameworks are optimized for first-order differentiation only. We refer to, for example, Kelly et al. (2020), for a discussion on higher-order differentiation in the context of a deep learning framework. To compute it efficiently, we manually implement forward-mode higher-order automatic differentiation for the operations needed in our context (described in Appendix C). A more efficient and general approach is left for future work. Our code is optimized for GPU.

**Penalization on a toy example.**   For Figure 2, the RNN is taken with 32 hidden units and hyperbolic tangent activation. The data are 50 examples of spirals, sampled at 100 points and labeled $\pm 1$

according to their rotation direction. We do not use batching and the loss is taken as the cross entropy. It is trained for 200 epochs with Adam (Kingma and Ba, 2015) with an initial learning rate of 0.1. The learning rate is divided by 2 every 40 epochs. For the penalized RNN, the RKHS norm is truncated at $N = 3$ and the regularization parameter is selected at $\lambda = 0.1$. Earlier experiments show that this order of magnitude is sensible. We do not perform hyperparameter optimization since our goal is not to achieve high performance. The initial hidden state $h_0$ is learned (for simplicity of presentation, our theoretical results were written with $h_0 = 0$ but they extend to this case). The accuracy is computed on a test set of size 1000. We generate adversarial examples using 50 steps of projected gradient descent (following Bietti et al., 2019). The whole methodology (data generation + training) is repeated 20 times. The average training time on a Tesla V100 GPU for the RNN is 8.5 seconds and for the penalized RNN 12 seconds.

Figure 3 is obtained by selecting randomly one run among the 20 of Figure 2.

**Libraries.**  We use PyTorch (Paszke et al., 2019) as our overall framework, Signatory (Kidger and Lyons, 2021) to compute the signatures, and SciPy (Virtanen et al., 2020) for ODE integration. We use Sacred (Klaus Greff et al., 2017) for experiment management. The links and licences for the assets are given in the following table:

| Name | Homepage link | License |
|------|---------------|---------|
| PyTorch | GitHub repository | BSD-style License |
| Sacred | GitHub repository | MIT License |
| SciPy | GitHub repository | BSD 3-Clause "New" or "Revised" License |
| Signatory | GitHub repository | Apache License 2.0 |