# OpenReview forum: "Framing RNN as a kernel method: A neural ODE approach"
_NeurIPS.cc/2021/Conference — NeurIPS 2021 Oral_

### Official Review · Reviewer_aWgR · 2021-07-14

**Rating:** 7
**Confidence:** 4

**Summary:**

This paper presents an approach to interpret RNN (in the continuous-time setting) as a kernel method. The main idea is to write the RNN ODE as a controlled differential equation, and through a signature transform of the (time-augmented) input process it is shown that the RNN output can be viewed as a linear function of the signature. Hence, RNN viewed this way is now amenable to classical analysis from Kernel methods. In particular, a generalization bound is proved in which the complexity measure is based on the defined RKHS norm of the target. Moreover, the approach inspires some regularization strategies for RNNs and the paper explores some of this via toy examples.

**Main Review:**

This paper is very well written and studies an interesting topic of embedding RNN as a linear function(al) in an RKHS. I think this forms a nice connection between time series methods and functional analysis / kernel methods, and as far as I am aware this is novel. More broadly, this may be an interesting mathematical framework to study RNNs and variants. While the main generalization bound here follows from classical kernel methods analysis, I think the value of the paper lies in identification of RNN as a linear (but infinite dimensional) map, thus allowing one to bring in these tools. Thus, I tend to vote for acceptance of this paper. Below, I list some questions the authors may wish to address.

1. As I understand, the $1/T$ term in various error/generalization bounds come from the time-discretization. If this is indeed the case, then I think it is of value to write this error separately (as in Prop 1) and then simply bound the rest of the errors in continuous time. In this way, it is more clear that the continous-time representation of the RNN as a linear map is exact, not approximate, as it currently reads in Theorem 1 (Eq 11)
2. Can you discuss some computational limitations of the signature? To me, there appears to be a huge curse of dimensionality (with respect to both dimension $d$ and “time points” $k$)
3. Here all results seems to be considered on a fixed temporal interval (with $T$ being the number of discretization points of this fixed interval). However, most RNN issues arise when time interval is expanded. Can you discuss the scaling of these results (e.g. the generalization bound in Theorem 2) respect to the time horizon under consideration?

---

After rebuttal:

I thank the authors for responding to my questions. I think this is an interesting paper and recommend acceptance.

**Time Spent Reviewing:**

4

---

> ### Author Response · Authors · 2021-08-07
> **Response to the review**
>
> 1. Reformulating the generalization bound to remove the $1/T$ term is a good suggestion. We gave a lot of thought into the best way of presenting our results. Due to the limited number of pages, we believe that the priority should be given to results that apply to the standard RNN, and decided therefore to keep the formulation intact. However, we will better stress the distinction between the terms depending on the discretization (in $1/T$) and the terms depending on the properties of the underlying continuous-time process and the architecture of the model. In particular, we will clearly point out in the new version that if the continuous-time process is handled directly, then the term in $1/T$ disappears. Thank you!
>
> 2. We thank the reviewer for highlighting the important issue of the computation cost, which we will discuss in the final version.  We realize that we have not been clear about the fact that we do not need to explicitly compute signatures---and this is why we chose not to discuss its computational limitations. Indeed, we are mostly interested in the norm of the network in the RKHS, which is entirely determined by the network parameters (see the explicit expression of $\alpha$ line 206). However, we fully agree that the computational complexity of signatures is an interesting question. First, note that the constant $k$ in Definition 1 does not correspond to the number of time points but to a truncation degree of the signature. In practice, we only compute signatures truncated at a certain order $k$. With a straightforward algorithm, the complexity of computing the signature at order $k$ of a path of dimension $d$ sampled at $T$ time points is $\mathcal{O}(Td^k)$. It is therefore linear in the number of time points but exponential in the truncation order. However, the truncation order need not be large since the difference between the full signature and its truncated version decreases exponentially fast (more precisely, it is a $\mathcal{O}(((1+L)/2)^{(k+1)})$. We will add a detailed comment on this important issue in the revised version.
>
> 3. Finally, the context mentioned by the reviewer (expansion of the time interval) is a very interesting question. We stress that it is essential to distinguish between the number of time discretizations $T$ and the time interval, say $[0,U]$. While the interval, and thus $U$, is fixed beforehand, $T$ is determined by the length of the data. Throughout, we have assumed that $U=1$. However, we could have taken another interval, say $[0,U]$, instead of $[0,1]$. This makes the analysis more complicated and requires strengthening some assumptions, in particular on the constant $L$, and does not bring much added value. In fact, our ‘continuous’ interpretation of RNN implies the following point of view on the `long-term’ issues: as long as the path remains sufficiently smooth on the interval $[0,U]$, all the analysis should remain valid, even if $T$ is very large. However, if the assumption that the path is of bounded variation is no longer valid on $[0,U]$, then the results do not hold anymore. In other words, the important parameter is no longer the data length $T$ but the regularity of the underlying process. We fully agree with the reviewer that a study of this question is very interesting but would drastically increase the technicality of our paper, and therefore we chose to delay it to future works.

---

### Official Review · Reviewer_jy69 · 2021-07-16

**Rating:** 7
**Confidence:** 3

**Summary:**

The authors consider a continuous data/neural ode model for recurrent neural networks. They prove convergence of the discrete model to the continuum model as the length of input sequences increases (assuming that sequences are obtained by finer and finer discretizations of the same process). They then show that the neural ODE can be represented alternatively as a controlled ODE, and that a solution of the controlled ODE can be expressed in terms of the signature of the input process. This allows the authors to equate the RNN to a kernel method for a specific kernel. Finally, they use Rademacher complexity bounds for the RKHS to estimate the generalization properties of RNNs in terms of the length of the input sequence/fineness of discretization. They use the continuum model to design a penalty term which improves stability as seen in numerical experiments.

**Limitations And Societal Impact:**

the authors adequately addressed the limitations and potential negative societal impact of their work.

**Main Review:**

The work presented is undeniably technically demanding and bridges diverse fields in an interesting way. I am not an expert, but I believe the underlying mathematics to be correct. The applicability of the results, however, appears to be quite limited at this stage. The coefficient bound in equation (10) is fairly severe for high-dimensional inputs. I am not sure how many interesting functions can be represented with such small weights - could the authors comment on the requirement?

While the article has definite merit, I believe that it presently falls short of inclusion in NeurIPS. Given the technical complexity and unsatisfying results, a more mathematically minded audience might appreciate this work more. It is undeniably an interesting direction of research, and I believe that a more mature contribution in this direction could be a valuable contribution in the future.

A few additional remarks and questions:

* The total variation of the process $X_t$ is bounded by $L<1$. Sequence data usually passes through several unit vectors. Could the authors comment more clearly on how the results scale with different upper bounds?
* l 61: Functions in $BV([0,1])$ are not generally continuous. Please adapt the notation e.g.\ $BV^c$ or similar (or relax the continuity requirement). I do believe that continuous functions form a closed subspace of $BV$, though.
* l 68: Subspace is usually used in the sense of `linear subspace' in this context. `Subset' might be a better term.
* l77/eq (2):  This is a very special case. If $\sigma$ is the logistic function (as suggested below), then $h$ is automatically monotone increasing coordinatewise since $\sigma>0$. A two-layer network needs an outer layer weight (at least for signed activation functions). This could easily be fixed by including a diagonal matrix of varying signs, for example, which does not change the norm bounds and does not require major changes.
* l 103: $M$ should be defined in the next line, where it is used (or the phrasing should be changed).
* l 144: missing subscript $[0,1]$.
* eq (10): The condition $\|W\|_F < \frac{Const}d$ is fairly severe when the input dimension $d$ is high, as the matrix becomes very large and the Frobenius norm sums over all coefficients. It is violated for any common initialization scheme. Could the authors discuss the bound and whether a weaker norm might suffice, or whether the tiny `radius of convergence' is in fact to be expected? The numerical results seem to suggest broader applicability.
* Appendix A.2: The Picard-Lindeloeff theorem should be considered standard. It does not need to be reproved here.
* l 666: In the last equation, the equality should be $= \frac{L+1}{L-1} \leq \frac{2}{1-L}$, I believe
* l 756: Infinity norm (rather than infinite norm)

**Time Spent Reviewing:**

4

---

> ### Author Response · Authors · 2021-08-07
> **Response to the review**
>
> We thank the reviewer for appreciating our research direction. It is clear that the present contribution is of a theoretical nature and we made our best in order to have clear and clean mathematical statements in this important domain. We have released open-source code to encourage future research in this direction and will be happy to see new contributions.
>
> The remark about the bound on the norm of the weights is very interesting. First, it should be noted that $d$ remains small in many applications involving temporal data (for example, $d$ is typically equal to 1 for univariate finance time series, 2 for character recognition, etc.). Moreover, this condition is merely a consequence of many technical steps and we believe that it could be improved (for example taking an upper bound sharper than the supremum upper bound on line 677), but at the price of additional messy technicalities and complexity for the reader. Therefore, this should not limit practical applications while a proof without this term remains difficult. Regarding the norm, we have the Frobenius norm for simplicity, but it is clear that using an ad hoc norm could improve the results, but again at the cost of increased complexity. All in all, we tried to provide precise results while keeping the article readable. We will explicitly mention it in the final version.
>
> Regarding the neural network model (line 77), we would like to clarify that we do not assume anything on the sign of sigma, and we take both the logistic function and the hyperbolic tangent (componentwise) functions as examples. Monotonic activation functions are standard in neural networks, and can emulate increasing or decreasing functions of their inputs (since the sign of $U$, $V$, $b$ in equation 2 is free). Furthermore, we would like to clarify that we do have an explicit outer layer weight in our model through the function $\psi$ (line 74), which is more general than the varying sign suggested by the Reviewer. See Example 4 for instance, where we explicitly add $\psi$ among the parameters. Hence we believe our setup to be fairly general. We will reformulate to make this clearer.
>
> Regarding the Picard-Lindelof theorem, we use here a non-standard version, with lightened assumptions: we consider a controlled differential equation instead of an ordinary one, and no assumption is made on the magnitude of the Lipschitz constant (whereas in the literature it is often chosen lesser than 1). The statement of the theorem comes from Lyons et al (2007) but we could not find the proof anywhere in the literature, and therefore chose to include it here for the sake of completeness.
>
> The bound on the total variation of the process ($L<1$) is necessary to ensure that signature coefficients do not explode, that is, to ensure the bound of Proposition 3. This is then necessary to have convergence of the Taylor expansion. In practice, this is a custom preprocessing step: we divide every sequence data by, for example, twice its total variation.
>
> Regarding other minor remarks of the reviewer:
> + Thanks for the remark on $BV([0, 1])$, we will change the notation in the final version.
> + Regarding the computation of l. 666: thank you for the detailed review! We checked and believe that our equality is correct (beware that the sum begins at $k=0$, your result would be valid if the sum began at $k=1$).

---

### Official Review · Reviewer_8hpD · 2021-07-18

**Rating:** 7
**Confidence:** 3

**Summary:**

This paper provides a rigorous framework to connect learning with RNNs and kernel methods, exploiting the idea that the output of a RNN can be viewed as a linear function of the signature of the input sequence in the continuous-time setting. Working within the framework (embedding the RNN into a RKHS), generalization bounds and stability guarantees are provided. Experiments are also provided to support the theory.

**Ethical Concerns:**

None.

**Limitations And Societal Impact:**

No, in particular no mention of the potential negative societal impacts.

**Main Review:**

The main strength of the paper lies in the rigorous analysis of the connection between RNNs and kernel methods in the continuous-time setting, which itself constitute a valuable contribution. The paper is easy to read and the results seem to be technically correct. While improvements in some areas (particularly, the experimental parts) could be made, overall it is a good paper and I am inclined to accept.

Minor comments:
(1) It seems like the embedding of the input sequence into a continuous path for use in the framework could play quite an important role. More comments/results on the effects of the embedding procedures (in terms of the discretization error incurred when discretizing a continuous-time input process) would strengthen the paper.
(2) Could one extend Theorem 2 to cover multi-class classification tasks?

**Time Spent Reviewing:**

2 hours

---

> ### Author Response · Authors · 2021-08-07
> **Response to the review**
>
> 1. Regarding the question about embedding, we stress that the framework we consider consists in deriving generalization bounds for continuous-time data. In practice, since we only have access to a discretized version of the process, we end up with a $1/T$ factor in Theorems 2 and 3, corresponding to a simple discretization procedure, with a $T$-step regular discretization. Therefore, all our bounds exhibit a clear distinction between a term depending on the discretization (in $1/T$) and a term depending on the properties of the underlying continuous-time process and the architecture of the model. Following your suggestion, we will stress out in the final version that the term $1/T$ comes only from the discretization procedure. This term could probably be improved by taking another data-dependent discretization scheme, although we feel this is out of scope of this paper, since it would greatly increase the technicality. Note also that computing the signature in practice requires a piecewise linear embedding of the input sequence, but we do not use this in the present article.
>
> 2. Extending Theorem 2 to multi-class classification tasks is a good suggestion. The method can be adapted without significant effort to the multi-class classification task (with appropriate loss functions). We will add a comment in the paper.

---

### Official Review · Reviewer_jQAU · 2021-07-23

**Rating:** 8
**Confidence:** 3

**Summary:**

The authors extend results regarding the application of controlled differential equations and rough paths theory, from 'feed-forward' architectures to recurrent neural networks.
The resulting analysis allows them to define an RKHS over such models, measure their capacity and bound their generalization.

**Main Review:**

I found this submission extremely interesting though quite technically challenging.

Theorems 2 & 3 are of great relevance to the entire community, and I would like to offer some minor suggestions:
1. I cannot find a clear definition of $\sigma^{(k)}$ in the main text. Please move the definition from the appendix into the main text.
2. An explicit comparison with existing generalization bounds for RNNs would make the submission more accessible. Moreover, though this is not imperative, it would be interesting to see an empirical comparison of the derived bounds with those found in the cited references.

I have a technical questions that I would like the authors to address.
For an ordinary differential equation rescaling $t \in \[0, T\]$ to $\tilde t \in \[0, 1\]$ requires rescaling appropriately the right hands side of the differential equation.
Does this imply that $K_f$ implicitly has such a dependence on $T$?

**Time Spent Reviewing:**

8

---

> ### Author Response · Authors · 2021-08-07
> **Response to the review**
>
> By $\sigma^{(k)}$, we mean the $k$-th derivative of $\sigma$. This will be clarified in the final version.
>
> An explicit comparison with existing bounds for RNNs is a very relevant suggestion. The paragraph below Theorem 2 already provides a brief comparison with existing generalization bounds. We stress that such a comparison is not straightforward since the generalization bounds found in the literature usually rely on assumptions different from ours and are architecture-specific, whereas our bound is quite generic. We will nevertheless extend this paragraph, and provide some numerical comparison of our bounds with those in the references.
>
> The question about rescaling is a very good remark, and we will clarify this point in the new version. First, it is important to note that one should distinguish between the number of time discretizations $T$ and the time interval $[0, U]$. While the interval, and thus $U$, is fixed beforehand, $T$ is determined by the length of the data. Throughout the paper we have assumed that $U=1$. However, we could have taken another interval, say $[0, U]$, instead of $[0,1]$, but then it makes the analysis more complicated (bounding the norm of the signature in Proposition 3) and does not bring much added value. A renormalization from $[0, U]$ to $[0, 1]$ would indeed impact some constants, and certainly $K_f$. Note however that there is no dependency of all the constants (including $K_f$) on the time discretization $T$.

---

### Author Response · Authors · 2021-08-07
**Thanks for the reviews**

Dear reviewers,

We warmly thank you for your time and relevant comments, which will help us improve our work. If accepted, we intend to take into account your suggestions, making use of the additional page.

We answer the specifics of questions pointed out by the reviewers in individual responses. We do not answer all minor remarks (such as style and typos), but they are well noted. Thank you!

Sincerely,

The authors

---

### Decision · Program_Chairs · 2021-09-27

**Decision:**

Accept (Oral)

**Comment:**

The paper establishes rigorous connection between (continuous time) RNNs and kernel methods. This paper brings a novel theoretical tool to the theoretical analysis of RNNs driven by a finite duration input. They also provide generalization properties and design a penalty term which improves stability as seen in numerical experiments. For the final version, please make sure to take into account all discussions from the rebuttal process.